

# New insights on resource stoichiometry: assessing availability of carbon, nitrogen and phosphorus to bacterioplankton

Ana. R. A. Soares[1], Ann-Kristin Bergström[2], Ryan A. Sponseller[2], Joanna M. Moberg[1], Reiner Giesler[4], Emma S. Kritzberg[3], Mats Jansson[2], Martin Berggren[1]

5  [1]Department of Physical Geography and Ecosystem Science, Lund University, Lund, SE-223 62, Sweden
[2]Department of Ecology and Environmental Science, Umeå University, Umeå, SE-901 87, Sweden
[3]Department of Biology/Aquatic Ecology, Lund University, Lund, SE-223 62, Sweden
[4]Climate Impacts Research Centre, Abisko, SE-981 07, Sweden

10  *Correspondence to*: Ana R. A. Soares (ana.soares@nateko.lu.se)

**Abstract.** Boreal lake and river ecosystems receive large quantities of organic nutrients and carbon (C) from their catchments. How bacterioplankton respond to these inputs is not well understood, in part because we base our understanding and predictions on 'total pools', yet we know little about the stoichiometry of bioavailable elements within the organic matter. We designed bioassays with the purpose to exhaust the pools of readily bioavailable dissolved organic carbon (BDOC), bioavailable dissolved nitrogen (BDN) and bioavailable dissolved phosphorus (BDP) as fast as possible. Applying the method in four boreal lakes at base flow conditions yielded concentrations of bioavailable resources that ranged from 105-693 µg C L$^{-1}$ for BDOC (2 % of total DOC), 24-288 µg N L$^{-1}$ for BDN (31 % of total dissolved nitrogen) and 0.2-17 µg P L$^{-1}$ for BDP (49 % of total dissolved phosphorus). Thus, relative bioavailability increased from carbon (C) to nitrogen (N) to phosphorus (P). We show that the main part of bioavailable nutrient resources is organic, representing 80 % of BDN and 61% of BDP. In addition, we demonstrate that total C : N and C : P ratios are as much as 13-fold higher than C : N and C : P ratios for bioavailable resource fractions. Further, by applying additional bioavailability measurements to seven widely distributed rivers, we provide support for a general pattern of relatively high bioavailability of P and N in relation to C. Altogether, our findings underscore the role of C as limiting factor for bacterial growth in boreal C-rich freshwaters, and suggest that these ecosystems are very sensitive to increased input of bioavailable DOC.

## 1 Introduction

Nutrient regulation of freshwater plankton productivity is central to the response of river and lake ecosystems to land use and climate change. By controlling phytoplankton primary production (PP) and bacterioplankton secondary production (BP), phosphorus (P) and nitrogen (N) are the two key macronutrients shaping aquatic ecosystems, with consequences for food web structure, biodiversity, and biogeochemical cycles (Jones, 1998). In addition to these nutrients, the supply of dissolved organic carbon has strong effects on ecosystem functioning by fueling BP and bacterial-based heterotrophic food chains (Dillon and Molot, 2005;Karlsson et al., 2012;Tranvik, 1998). While nutrient availability can be influenced by internal lake




processes, the regulation of PP and BP in the majority of lakes worldwide is constrained by loading of inorganic and organic resources from the surrounding terrestrial landscape (Wetzel, 2001). In brown-water boreal lakes, nutrients bound to dissolved organic matter (DOM) (e.g., humic substances) often dominate inputs (Jansson, 1998). In such systems, terrestrial nutrient support of BP is of particular ecological and biogeochemical importance, as heterotrophic processes often greatly

exceed autotrophy (Jansson et al., 2000).

While the importance of nutrient availability at the ecosystem level is evident, characterizations of the actual proportion of terrestrially-derived resources that can be readily used by aquatic microorganisms are difficult and attempts are rare. A variable fraction of C, N, and P of terrestrial origin is chemically bound in organic molecules that are typically too large to be directly taken up by microbes (Battin et al., 2008). The nature of the covalent bonds and the structure of organic

compounds that hold N and P also differentially influence the bioavailability and turnover of associated nutrients (Vitousek et al., 2002). Such complexity makes it difficult to predict the potential for bacterial usage of these resources at ecologically relevant-scales (Bronk et al., 2007;Berggren et al., 2015;Helton et al., 2015). It is generally thought that the major fraction of dissolved organic carbon (DOC) originating from terrestrial soils is recalcitrant, yet bioavailability estimates from different lakes suggest that a variable proportion of DOC can be used by bacteria (6-14 %; Tranvik, 1988). For dissolved organic

nitrogen (DON), a summary of published assays suggests that anywhere from 2-75 % of the organic N pool may be bioavailable (Pellerin et al., 2006), with a range of 19-28 % reported for boreal streams during base flow (Stepanauskas et al., 1999). While less studied, P bioavailability appears to be equally variable over space and time (Muscarella et al., 2014). For example, it has been shown that seasonal concentrations of bioavailable P ranged from 1 to 14 µg P L$^{-1}$ in boreal headwater streams, representing from < 5% to nearly 50% of the total P pool (Jansson et al., 2012). Most studies on nutrient

availability conducted in humic-rich waters have neglected this variability in bioavailability, focusing on either on total inputs (i.e., total N or total P) or on the turnover of specific fractions assumed to be bioavailable (e.g., dissolved inorganic nitrogen, DIN; molybdate reactive phosphorus, MRP). However, inorganic fractions may constitute only a small part of the total nutrient pools and can underestimate resource bioavailability in organic-rich waters with large pools of labile DON or dissolved organic phosphorus (DOP; Seitzinger et al., 2002).

Pitfalls of assuming availability and resource limitation from total pools or inorganic fractions to bacterioplankton have prompted the suggestion that standardized bioavailability assays (re-growth bioassays) should be incorporated into the analytical toolbox of researchers (Lewis, 2011). These are operationally defined bioavailability measurements in which an inoculum is added to a sterile solution and the bacterial biomass is allowed to grow during a standardized incubation at a determinate temperature. This re-growth response is used to assess how much resources were consumed during the

incubation, which is a measure of bioavailability (sensu Berggren et al., 2015).

Unfortunately results from the few different studies addressing bioavailable resource shares for bacterioplankton are difficult to compare since different methodological approaches are used (Berggren et al., 2015). For instance, studies of DOC bioavailability have used methods that differ in terms of incubation length, temperature, inorganic nutrient concentrations, as well as in the approach used to inoculate samples with microbial communities (del Giorgio and Davis, 2003). Similarly, as





different techniques and assumptions have been applied to assess nutrient availability, results for N and P differ among studies and are generally not comparable as they often reflect variation in experimental factors rather than in the intrinsic molecular properties of the nutrients themselves. Thus, a standard and comparable method that can tackle the bioavailability of multiple elements to bacterioplankton is still missing.

Previous attempts to measure nutrient bioavailability of multiple elements have mainly been performed over very long time-scales (most data from 100-day incubations; see data review by Lonborg and Anton Alvarez-Salgado, 2012) and do not represent the pool that is immediately available for consumption. These assays have not been based on re-growth, but on long-term changes in bulk nutrient concentrations in solution (Lonborg and Anton Alvarez-Salgado, 2012). To move the nutrient stoichiometry field forward, a promising option is to measure the uptake of nutrients through growth bioassays

conducted at more ecologically relevant timescales, i.e. only long enough to exhaust the readily bioavailable nutrient pool during a few single days (sensu Berggren et al., 2015). Although growth bioassays have previously been applied to calculate bioavailability of single elements (Stepanauskas et al., 2002;Jansson et al., 2012;Stepanauskas et al., 2000), no such efforts to date have quantified the bioavailability of more than two elements simultaneously, so that the relative availability can be directly compared. In a recent review on bioavailability (Berggren et al., 2015), it was additionally suggested that nutrient

bioavailability (as a fraction of the total pool) actually may increase from C to N and N to P in DOM-rich systems. While this hypothesis is generally consistent with our understanding of resource use in soils (Vitousek et al. 2002), it has yet to be accurately tested in surface waters.

In this study, we designed bioassays with the purpose to rapidly exhaust the pools of readily available organic C, N and P, accessible to bacterioplankton in DOM-rich lakes. The bioassays were designed such that most of the nutrients were used

within three days, although we measured the cumulative nutrient use during up to seven days. We first calibrated our method by detecting the response (leucine incorporation) of nutrient-starved bacteria to known added amounts of bioavailable resources. We further validated this bacterial response through comparison with common methods to detect bioavailability: lability incubations for DOC bioavailability (del Giorgio and Cole, 1998), growth bioassays with N starved bacteria for N bioavailability (Stepanauskas et al., 2000) and measuring P content in bacterial growth cultures harvested on filter (Jansson

et al., 2012). Specifically, by using this new bacterioplankton growth bioassay our study aimed to ask: 1) How does the relative total bioavailability in DOM-rich surface waters differ between the elements, i.e. bioavailable dissolved organic carbon (BDOC) out of total DOC, bioavailable dissolved nitrogen (BDN) out of total N and bioavailable dissolved phosphorus (BDP) out of total P, respectively, and do these shares vary seasonally?; 2) Are the organic bioavailable N and P pools larger than the corresponding inorganic pools?; 3) Does the use of total C:N, C:P and N:P yield ratios that are higher

than the actual ratios between bioavailable C:N, C:P and N:P. This was tested by performing bacterial growth bioassays on four boreal lakes in northern Sweden with high DOM concentrations. In addition, we applied a simplified version of our new method to assess broad patterns in nutrient bioavailability across a larger cross-regional scale and climate gradient that compromises seven river systems with variable DOM concentrations.




## 2 Methods

### 2.1 Study area and sampling

We studied four lakes in northern boreal Sweden: Övre Björntjärnen, Lillsjöliden, Struptjärnen and Stortjärnen. All lakes are unproductive brown-water systems of similar size and morphology (Table 1). Lake catchments are dominated by coniferous

forest (Scots Pine; *Pinus sylvestris* and Norway spruce; *Picea abies*) and wetlands (mires) in different proportions. The lakes are closely co-located (maximum distance 75 km) and influenced by similar climatic conditions. Average annual temperature, precipitation and runoff in this area are approximately 1.8 °C, 614 mm, and 311 mm, respectively (from 1981-2010; Laudon et al., 2013). Lake surface ice coverage extends from November to May; stratification occurs during late May/early June and mixing occurs after mid-September.

In addition to these lakes, we also sampled the outlet of seven Swedish rivers (Lyckebeån, Helge å, Nyköpingsån, Motala Ström, Torne älv, Töre älv, Öre älv) that drain into the Baltic Sea. River catchments are located between latitudes 55°N and 65°N, falling along a 1300 km north-south gradient, spanning a range of drainage areas of 440-34441 km$^2$, and with DOC concentrations from 5.6 to 23 mg L$^{-1}$. These rivers drain very different terrestrial environments from mountains, forests, and wetlands in the north to catchments with a significant fraction of agricultural land and urban development in the south

(Sponseller et al., 2014). In addition, these systems are influenced by different climates, from sub-arctic in the north to temperate in the south. From north to south, average temperature, precipitation and runoff respectively span from 1-8°C, 631-824 mm, and 34-450 m$^3$/s (for 1999-2013; Swedish Meteorological and Hydrological Institute, SMHI).

Lake samples (2 L) were collected from 0.5 m depth at seven dates from September 2012 to September 2014 (Table 2). Samples were stored in acid washed 2 L high-density polyethylene bottles and 4 L low-density polyethylene cubitainers

(Thermo Scientific) in the dark at approximately 1 °C until arrival at the laboratory. River sampling was conducted once at the outlet of each river between June to July 2013 at 0.3 m depth, in the middle of the river or 7 m from the shore.

### 2.2 Determination of bioavailable C, N and P

To determine concentrations of BDOC, BDN and BDP we conducted growth bioassays in which limitation of either C, N or P was strongly induced by adding different combinations of bacterial growth media. Our growth bioassays were designed so

that resource use efficiency was at its maximum and bacterial production would occur mostly within three days from the beginning of the experiment. The bacterial response to those bioassays was measured by leucine incorporation (Kirchman et al., 1985). The amount of leucine incorporated in each bioassay was then converted into concentrations of bioavailable resource based on experimentally determined standard growth curves (see detailed description below).

Bioassays were prepared immediately after or at latest within one to two weeks after sampling. To ensure proper

conservation of the samples prior to the experiment, they were immediately filtered (Whatman GF/F) and stored in a climate controlled chamber at a temperature close to 1 °C. At the initiation of the experiment, 500 ml of each lake and river water sample was again filtered at 0.2 μm (suporCap 100, Gelman Sciences) and placed in a 1000 ml Erlenmeyer flask. All





bioassay samples were then inoculated with a standard bacterial community 2 % (v/v), which consisted of a mixture of fresh unfiltered stream and river water from the nearby the field sites sampled at one occasion. The water was amended 5% (v/v) with a modified (excluding C, N and P) bacterial medium ("L16"; Lindström, 1991) rich in micro-nutrients, trace metals and vitamins required for bacterial growth. The sample was then divided into three sub-volumes to which strong limitation of

either C, N or P was induced by adding appropriate combinations of nutrients. C limitation was induced by adding N as $NH_4NO_3$ (final concentration 2000 µg N $L^{-1}$) and P as $Na_2HPO_4$ (200 µg P $L^{-1}$). N limiting conditions were created by adding C as $C_6H_{12}O_6$ (20000 µg C $L^{-1}$) and P as $Na_2HPO_4$ (200 µg P $L^{-1}$). P limiting conditions were created by adding C as $C_6H_{12}O_6$ (20000 µg C $L^{-1}$) and N as $NH_4NO_3$ (2000 µg N $L^{-1}$). Samples were then transferred into 1.5-ml Eppendorf tubes, which were incubated in the dark at the standard temperature of 20°C, which is the most broadly applied temperature in

bioavailability assessments of the literature (del Giorgio and Davis, 2003). For each bioassay incubation, leucine incorporation was measured at six time points (after 0, 1, 2, 2, 3 and 7 days) on five replicate samples each time. The inoculum added to our bioassays represents an unknown addition of bioavailable C, N and P. To ensure that the amount of resource added through inoculation was insignificant, we analyzed five control bioassay replicates in which the only source of C, N or P was the amount of resource contained in the inoculum and thus the lake sample was replaced by Mili-Q water.

All such control bioassays resulted in low amounts of leucine uptake (Fig. 1), which was then used to correct our estimates of resource bioavailability through subtraction.

To create standard curves for bacterial growth per unit limiting nutrient, sampled lake water from September 2012 was used to perform a bioassay following the approach described above but with varying concentration of target elements. For example, to a sub-volume that was induced to be C limited, $C_6H_{12}O_6$ was added to final concentrations of 330, 660, 1000,

1330, and 1500 µg C $L^{-1}$ respectively. The response to each concentration was measured on one to triplicate samples and was used to construct the standard curve. The same procedure was applied to produce standard curves for N and P limited assays. $NH_4NO_3$ was added to concentrations of 105, 133, 205, 305, 405 µg N $L^{-1}$, and $Na_2HPO_4$ was added to concentrations of 15.5, 18.8, 20.5, 30.5, 40.5 µg P $L^{-1}$. Standard curves for the rivers were based on the same approach but bacterial responses to each concentration were recorded one time.

Integrated (cumulative) amounts of leucine incorporated by bacteria during lake or river bioassays over seven days were converted to concentrations of bioavailable element based on the slopes of the standard growth curves of either rivers or lakes, which describe how much leucine was incorporated per unit of bioavailable limiting element. For this conversion, the amount of incorporated leucine (given in nmol of leucine $L^{-1}$ per seven days) during each bioassay was divided by the slope of the standard growth curve (nmol of leucine $L^{-1}$ per mg of bioavailable nutrient $L^{-1}$ for seven days). The resulting quotient

represents the total amount of bioavailable nutrient taken up by bacterioplankton (mg $L^{-1}$ for seven days).

### 2.3 Leucine incorporation

Measurements of protein synthesis were done using the method described by Smith and Azam (1992) and modified by Karlsson et al. (2002). Accordingly, $^3$H-leucine was added to sample water in Eppendorf tubes (specific activity varied



between 60.5-115.8 Ci mmol$^{-1}$, Perkin Elmer) to a final concentration of 30-100 nmol L$^{-1}$. Additions of $^3$H-leucine were dependent on bacterial activity tests performed prior to the experiments where different concentrations of $^3$H-leucine identified the isotope saturation levels. Triplicate measurements were taken after 24 h, 48 h (we obtained six replicates at this time point), 72 h, 96 h and 168 h. Leucine incorporation into protein was determined by incubation for 1 h in the dark at 20 ˚C and incubations were terminated with trichloroacetic acid (TCA) additions of 5 % (w/v). A bacterial pellet was formed by centrifugation for 10 min at 14 000 rpm. The bacterial pellet was rinsed with 5 % TCA. After addition of 1.2 mL of scintillation cocktail (PerkinElmer) radioactivity was measured on a Wallac WinSpectral 1414 Scintillation counter (PerkinElmer). Incorporation of $^3$H-leucine was calculated using an intracellular dilution factor of 2 (Smith and Azam, 1992). Leucine incorporation measurements were integrated for the six time points and summed into a single value that represented the total amount of leucine incorporated for the seven day period. Lastly, at time point 96 h, an extra vial was collected and used as a blank, pre-treated with TCA 5% (w/v), followed by addition of leucine at a final concentration of 30 nmol L$^{-1}$.

## 2.4 Validation

We validated the bacterial responses (leucine uptake) to added amounts of BDOC, BDN and BDP by measuring the leucine uptake per unit ambient bioavailable resource measured with alternative methods. An alternative estimate of BDOC was obtained from measuring bacterial respiration (BR) during a lability incubation, which has been often applied in aquatic research (del Giorgio and Cole, 1998;Jansson et al., 2000). BR was determined by assessing decreases in dissolved oxygen concentrations from bioassays from lakes (n=13) and rivers (n=8). Sample water was prepared in parallel with, and in the same way as, the C bioassays described above. Volumes of 0.5 L were added to glass incubation bottles (in duplicate) which had sensors spots affixed to the inside surface. Oxygen concentrations were measured in the dark every 5 min for up to seven days with a FIBOX 3 (PreSens) that took optical readings from the outside of bioassay bottles. Estimates of BR were calculated from the averaged consumption of dissolved oxygen from the duplicate bottles by assuming a respiratory quotient of 1, which is a conservative value for unproductive lakes (Berggren et al., 2012).

Bioavailable N was assessed using an alternative method described by Stepanauskas et al. (2000) by counting the cells produced in growth bioassays with N starved bacteria. For this test, two aliquots of 30 ml were used for bioassays and one of them was amended with N-NH$_4$NO$_3$ to a final concentration of 0.405 mg N L$^{-1}$. Both incubations were performed at 20 ˚C degrees in the dark. Bacterial biomass was determined at the start of the incubation (t=0) and after three days (t=3) after the bacterial growth had peaked (Fig. 1). Bacterial samples were fixed with 3 % (v/v) glutaraldehyde and kept at 5 ˚C until analysis. Analyses of bacterial cells were conducted on a flow cytometer (FACScan, Becton Dickinson) on samples stained with SYTO 13 and run with addition of beads as internal standard according to del Giorgio et al. (1996), using CellQuest Pro software. Bacterial cells were distinguished based on green fluorescence intensity and side scatter signals. Total bacterial abundance was calculated as the sum of the populations that were distinguished in the cytograms. The N content per bacterial cell was determined by dividing the amount of N added to the amended aliquot by the difference in bacterial





abundance between the N amended and the unamended aliquot. To obtain BDN the calculated average N content per cell was multiplied by the number of bacterial cells that were produced in the bioassay without addition.

To validate our estimates of P bioavailability, we extracted data from Jansson et al. (2012) where bioavailability was assessed by an alternative approach from two northern Swedish streams from late April to late October, and in addition

cumulative leucine incorporation during the bioassays was measured through the method described in this study. This alternative approach was used to determine concentrations of bioavailable P as the difference in the particulate P (retained on a nominal cutoff of 0.2 μm filters, Supor AcroPak 200, Pall Corporation) at the end and in the beginning of a four-day experiment, which should correspond to the amount of P taken up by bacteria during the incubation period.

## 2.5 Analytical methods and calculations

Lake ambient water chemistry was analyzed at the department of Ecology and Environmental Science at Umeå University. Sample water for determination of DOC and TDN was filtered through a pre-ignited (400 ˚C, 3 h) acid-rinsed Whatman GF/F filters. The filtered water was acidified with 1.2 M HCl and analyzed for DOC using a HACH-IL 550 TOC-TN. Filtered sample was analyzed for TDN also using a HACH-IL 550 TOC-TN, while determination of nitrate ($NO_3^-$) and ammonium ($NH_4^+$) was done according to the International Organization for Standardization (ISO) 13395-1996.

Concentration of phosphate ($PO_4$-P, assumed to be represented by soluble reactive P) was determined from filtrates (GF/F) of water samples using the molybdate blue method Murphy and Riley (1962) and total phosphorus (TP) determined after oxidative hydrolysis with potassium persulfate (ISO 15861-1).

River DOC samples were filtered through a Whatman GF/F filter into a pre-acid washed 40 ml amber borosilicate vial, filled to the brim and tightly closed with silicon septa screw caps. Samples were kept cold in the fridge until analysis which took

place at the G.G. Hatch Stable Isotope Laboratory, University of Ottawa. River samples for determination of (total dissolved nitrogen) TDN, $NO_3^-$, $NH_4^+$, TP and $PO_4$-P were frozen until analyses at the Evolutionary Biology Center, Uppsala University following standard methods.

Our results provided estimates of total bioavailable resource pool. To calculate shares of bioavailable DON (BDON) and bioavailable DOP (BDOP), we subtracted the inorganic pools of DIN ($NO_3^-$, $NH_4^+$) and $PO_4$-P from the respective total

bioavailable pools. Nutrient ratios were calculated in molar. We further calculated inorganic nutrient ratios of DIN to $PO_4$-P (DIN : $PO_4$-P).

## 2.6 Statistical analyses

Standard curves were fit by linear regressions using JMP 10 (SAS). Differences between the slopes of standard curves for each nutrient across lakes were tested by one-way analysis of variance (ANOVA, $p < 0.05$) in SPSS 22.0 (SPSS Inc.,

Chicago, IL, U.S.A.). Since there were no statistical differences between slopes obtained for each respective resource (ANOVA, $p > 0.44$, $n = 20$), slopes were averaged for each nutrient across lakes. Differences between bioavailable resources results across lakes and for each lake across time were tested using the Kruskal-Wallis H test and Dunn's post-hoc test ($p <$



0.05) in SPSS. Differences between total and bioavailable resource ratios for the lakes were tested with dependent t-test ($p = 0.05$) in SPSS. Previous work suggests that at higher DOM concentrations there is a greater discrepancy between bioavailable and total DOM fractions (Berggren et al., 2015). We therefore pooled the seven different rivers into two categories according to their DOC concentrations (3.7-23.0 mg C L$^{-1}$); this resulted in an ensemble of three rivers which had

a DOC concentration higher than 10 mg C L$^{-1}$ (rivers$_{>10\ mg\ C\ L}$$^{-1}$) and four rivers that had a DOC concentration lower than 10 mg C L$^{-1}$ (rivers$_{<10\ mg\ C\ L}$$^{-1}$). Differences between total and bioavailable river nutrient ratios for the two groups were tested with dependent t-test ($p = 0.05$) in SPSS.

## 3 Results

The rate of leucine incorporation increased over time in most bioassays until day 2 (t = 2), before gradually decreasing until day 7 (t = 7; Fig. 1). In the bioassays that were performed with resource additions, the accumulated leucine incorporation over the 7-day period was proportional to the concentrations of bioavailable resource added (Fig. 2). The results rendered an average linear relationship describing amounts of leucine incorporated per bioavailable C, N and P (Fig. 2).

Bioavailable resource concentration spanned from 104-692 µg C L$^{-1}$, 23-287 µg N L$^{-1}$ and 0-16 µg P L$^{-1}$ (Table 2).

Concentrations of BDOC did not differ among lakes (ANOVA, $p > 0.61$, $n = 130$). By contrast, the four lakes did vary in terms of average BDN and BDP (ANOVA, $p < 0.05$, $n = 130$). Lake Struptjärnen had on average the highest BDN and BDP concentrations (159 µg N L$^{-1}$ ± 111 SE and 8 µg P L$^{-1}$ ± 4 SE) and lake Lillsjöliden the lowest values (124 µg N L$^{-1}$ ± 97 SE and 5 µg P L$^{-1}$ ± 3 SE).

There was a significant difference in bioavailable resource concentrations over time across the lakes (ANOVA, $p < 0.05$, $n =$

30-35; Table 2). In general, concentrations of BDOC across the lakes were higher in October 2012 (mean 356 µg C L$^{-1}$ ± 84 SE) and lowest in August 2014 (mean 185 µg C L$^{-1}$ ± 59 SE), with a 33 % difference in BDOC between maximum and minimum values during the studied period. Concentrations of BDN tended to be high in September 2012 (mean of 236 µg N L$^{-1}$ ± 45 SE) and lowest in September 2014 (mean of 58 µg N L$^{-1}$ ± 42 SE) and varied 85 % between the maximum and the minimum concentration. Concentrations of BDP were highest in July 2014 (mean of 12 µg P L$^{-1}$ ± 3 SE) and lowest in

October 2012 (mean of 4 µg P L$^{-1}$ ± 3.6 SE) and varied approximately 83 % throughout the studied period. Average fractions of bioavailable resource relative to the total pool were lowest for C, highest for P, and intermediate for N (Fig.3). Organic forms were the major source of bioavailable resources to bacterioplankton and represented 80 % (± 13 SE) of the bioavailable N pool and 61 % (± 46 SE) of the bioavailable P pool (Fig. 3). The contribution of inorganic fractions was therefore more important for overall P bioavailability.

Molar nutrient ratios calculated for the total pool of nutrients were significantly higher than ratios calculated with basis on the bioavailable fraction (dependent t-test, $p < 0.05$, n = 26; Fig. 4). For example, the average ratio of total C : N was 55 (±9 SE) and was ca 13 times higher than C : N bioavailable ratio which averaged 4 (± 3 SE). Similarly, average C : P total ratio




was 4774 ($\pm$ 2135 SE) and was 12 times significantly higher than the average bioavailable C : P ratio 369 ($\pm$ 915 SE). However, there were no significant differences (dependent t-test, $p > 0.474$, $n = 26$) between total N : P ratios (average of 145 $\pm$ 386 SE) and bioavailable N : P ratios (average of 89 $\pm$ 44 SE), or between bioavailable N : P ratios and the DIN : $PO_4$-P ratio (mean of 29 $\pm$ 19 SE; dependent t-test, $p > 0.134$, $n = 26$).

The amounts of leucine incorporated per unit of bioavailable resource in our re-growth bioassays (as determined by the slopes in Fig. 2) was validated by extracting the same ratio from experiments performed using alternative bioassay methods (Fig. 5). The alternative bioassay methods were based on: 1) inferring BDOC from bacterial respiration; 2) calculating BDN from cell yields and; 3) analyzing BDP directly on the bacterial biomass (see methods). The growth responses (leucine incorporation) in our re-growth bioassays overlapped with the growth responses obtained from experiments using the
alternative methods. However, on average the growth response was slightly higher in our bioassays when compared to the alternative bioassays (Fig. 5).

For rivers, DOC appeared as the least bioavailable resource (in relation to the total pool) for both river groups: rivers$_{>10\,mg\,C\,L}$-1 and rivers$_{<10\,mg\,C\,L}$-1 (Table 3). In contrast, the BDN share was the most bioavailable with approximately half of the TN pool being bioavailable. Total nutrient ratios of C : N and C : P were statistically significantly higher (approximately 26 and 5-
fold respectively) than the respective bioavailable resource ratios for rivers$_{>10\,mg\,C\,L}$-1 (dependent t-test, $p < 0.05$, $n = 4$). We found no differences between total N : P ratio and bioavailable N : P ratios, neither between each of these and DIN : $PO_4$-P ratios for both rivers$_{>10\,mg\,C\,L}$-1 (dependent t-test, $p > 0.07$, $n = 4$) and rivers$_{<10\,mg\,C\,L}$-1 (dependent t-test, $p > 0.10$, $n = 3$).

## 4 Discussion

### 4.1 Resource bioavailability as driver of ecological patterns

Results from this study underscore ineffectiveness of total nutrient fractions as predictors of bioavailability in boreal freshwater ecosystems. Surprisingly, in these systems where absolute surface water DOC concentrations are large, C bioavailability was low and was the strongest limiting factor for heterotrophic aquatic production. This study not only reveals the pervasive control that C has on boreal heterotrophic aquatic production but also suggests that possible changes in C loading to the boreal water systems in the future may impact aquatic productivity and the turnover of nutrients. Northern
catchments are thought to be particularly sensitive to ongoing climate change (Tetzlaff et al., 2013) and this refined understanding of bioavailable resource stoichiometry may be essential to forecast and mitigate aquatic ecosystem responses to these and other anthropogenic pressures at high latitudes.

### 4.2 Bioavailable concentrations of DOC, TDN and TDP in lakes

Our estimates, which reflect the resource pool available to bacterioplankton at any point in time, supported our expectations
by showing that nutrient bioavailability (as percentage of the total pool), increased from BDOC to BDN and from BDN to BDP. The observed differences in N and P bioavailability match the overall trend reported for aquatic ecosystems in the





literature (Berggren et al. 2015) and are generally consistent with our understanding of how these elements are bound to organic matter. Organic N tends to form covalent bonds directly to C and may be physically and chemically protected within complex, organic compounds that are resistant to decay (Schulten and Schnitzer 1998). Liberating this N is linked to organic matter depolymerization and C mineralization (Schimel and Bennett 2004), requiring multiple exo-enzymatic steps that are

energetically expensive (Sinsabaugh and Follstad 2011). By contrast, organic P is more often associated with ester bonds (C-O-P) that can be cleaved in a single enzymatic step independent of C mineralization (McGill and Cole 1981). In addition, other forms of inorganic P (e.g., orthophosphate) may be only loosely bound and exchanging with iron-humic complexes (Jones 1998). These binding properties are thought to govern differences in the relative rates of N and P cycling in soils (Vitousek et al. 2002) and our results suggest that the same factors may shape the relative bioavailability of these resources

also in freshwater environments.

The method we describe here generated simultaneous bioavailability estimates for C, N, and P that were comparable to those from single element bioassays reported elsewhere. Absolute concentrations of BDOC (100-690 µg C L$^{-1}$) were within the range of reported values for cedar bog wetlands (12-408 µg C L$^{-1}$; Wiegner and Seitzinger, 2004) and were at the lower end of values reported for rivers (108-180 µg C L$^{-1}$; Wiegner et al., 2006). Concentrations of BDN (30 - 320 µg N L$^{-1}$) were in

agreement with bioavailable N concentrations reported for cedar bog wetlands (0-322 µg N L$^{-1}$;Wiegner and Seitzinger, 2004). BDP (0-16 µg P L$^{-1}$) was comparable to values from a recent study on headwater streams during low flow (1-14 µg P L$^{-1}$; Jansson et al., 2012). In addition, organic forms dominated the total bioavailable N and P pool (80 % and 61 % respectively) in our four lakes, and 27 and 36 % of these organic pools were bioavailable, respectively. These results are in line with previous estimates and show that a large fraction of DON is available to bacterioplankton in diverse limnic

systems, e.g. in Baltic Sea rivers (30 %; Stepanauskas et al., 2002), in eastern US rivers (23 %; Wiegner et al., 2006) and in cedar bog wetland streams (33 %; Wiegner and Seitzinger, 2004). Published estimates of the share of BDOP (bioavailable dissolved organic phosphorus) relative to the total DOP pool, varied from 33-60 % in Baltic Sea brackish waters (Nausch and Nausch, 2007). Thus, our results agree with the results from previous studies and together they emphasize the importance of organic nutrient fractions in systems rich in organic matter, and also the bacterioplankton capacity to take up

organic compounds.

Concentrations of BDOC, BDN and BDP varied seasonally in all lakes during the study period (Table 2). Major differences in BDOC were observed between mid-summer, when concentrations were lowest, and the end of the summer, when concentrations were high. Previous experimental work on boreal and arctic rivers has also shown minimal concentrations of BDOC during the summer season (Wickland et al., 2012). In addition, concentrations of BDOC tended to follow bulk DOC

concentrations in boreal freshwater systems as suggested in Søndergaard and Middelboe (1995). Because the design of our lake experiment controlled for most factors affecting bacterioplankton C uptake (i.e. temperature, bacterial communities, predation, hydrological conditions, inorganic nutrient concentrations, land-use differences; del Giorgio and Davis, 2003), the variation in the amount of BDOC was most likely coupled to seasonal temperature fluctuations which influence soil microbial activity and consequently the quality of the exported organic C to surface waters (Kalbitz et al., 2000;Carlson et



al., 2002). By contrast, patterns of BDP concentrations opposed those of BDOC (Table 2): specifically, BDP peaked in mid-summer (July) and declined in the autumn. It has been shown elsewhere that bioavailable P concentrations in boreal streams can be 2-10 times higher during summer than during autumn (Jansson et al., 2012). This may be due to the fact that low BDOC concentrations in forests soils during summer lead to reduced uptake (i.e. reduce biotic demand for P) and

consequently result in exports of DOM depleted in labile C and rich in bioavailable P (Jansson et al., 2012).

Our results also supported the prediction that the bioavailable ratios of C : N and C : P would be considerably lower than counterparts based on total pools. A major implication of these differences is that ratios based on total pools grossly overestimate actual C availability. When such differences are large, the elemental ratios based on total pools can lead to incorrect predictions of resource limitation (Berggren et al. 2015). For example, in a recent study of two temperate estuaries,

total resource stoichiometry predicted P limitation of bacterioplankton, while experimental evidence showed that C was the element constraining bacterial growth during base flow (Hitchcock and Mitrovic, 2013). Average DIN : $PO_4$-P ratios and particularly total TN : TP were however, closer to the average ratio of bioavailable TN : TP. Due to the high C recalcitrance, nutrient limitation predictions based on the ratio of total resource pools may be inadequate when C is included the ratio, but seem more promising when based on N and P.

Our results further show that while the median bulk stoichiometric ratio (3651C:71N:1P; Fig. 4) was 1-2 orders of magnitude higher than that expected from the Redfield ratio (106C:16N:1P; Anderson, 1995;Redfield, 1958), the median C : N : P of bioavailable resources (144C:29N:1P) was surprisingly comparable yet slightly above Redfield values (Fig. 4). There was, however, a wide variability in the bioavailable ratios among samples, which is consistent with another study that evaluated bacterial biomass stoichiometry over a large number of lakes and showed that, while elemental stoichiometry

varied among lakes in response to intrinsic and extrinsic factors, the overall mean ratio tended to converge with Redfield (Cotner et al., 2010). It should be pointed out here that the C:N:P content of cells does not necessarily represent the relative rates of supply that are required for these elements, particularly given that relative C incorporation into biomass can be highly variable (also called bacterial growth efficiency; BGE). Considering the need for carbon to fuel respiration and build biomass, actual uptake ratios of C : N and C : P must take into account the fact that BGE in boreal waters can vary with the

source and age of the terrestrial carbon from 0.06 to 0.50 (Berggren et al., 2007). Applying this reported possible range of BGE to our data suggests that the median bacterial demand for C : N and C : P varies between 7-58 and 290-2421, respectively. Considering that only a fraction of the bulk C, N and P was available for uptake (2 %, 31 % and 49 %; Fig. 3) the actual median C : N (3) and median C : P (166), was lower than these uptake ratios corrected for BGE, providing further support that C was limiting in all our samples.

**4.3 Broad-scale riverine BDOC, BDN and BDP patterns**

Broad-scale patterns of nutrient bioavailability at the river mouths did not differ between rivers$_{<10 \, mg \, C \, L}$$^{-1}$ and rivers$_{>10 \, mg \, C \, L}$$^{-1}$. Similar to what was observed in the lakes, DOC was the most recalcitrant nutrient considered. However, in contrast to our results from the lakes, TDN was the most bioavailable resource observed in the river mouths (Table 3). Although previous



studies suggest that temperature differences across catchments can influence C : N ratios in streams and rivers through effects on terrestrial ecosystem properties (e.g., vegetation type) and soil development (Sponseller et al., 2014), our results show a similar bioavailable resource stoichiometry at the outlet of all these rivers. Organic forms of N were a major source of bioavailability and dominated TDN, in agreement with estimates from other studies (Wiegner et al., 2006;Seitzinger and

Sanders, 1997;Stepanauskas et al., 2002). Significant differences between total and bioavailable C : N and C : P ratios occurred only in rivers$_{>10 \text{ mg C L}}$$^{-1}$. Whereas, both rivers$_{>10 \text{ mg C L}}$$^{-1}$ and rivers$_{<10 \text{ mg C L}}$$^{-1}$ had no differences between total N : P, bioavailable N : P and DIN : PO$_4$$^{3-}$ ratios. These results indicate that, similar to lake results, the use of bulk resource ratios misinterprets resource bioavailability and limitation when 1) C is part of the nutrient ratio and 2) there is a high concentration of DOC in the waters.

## 4.4 Measuring bioavailability of C, N and P with leucine incorporation

The linear relationships obtained from standard growth curves relating leucine incorporation to bioavailable resource concentrations showed that incorporation over a 7-day period was significantly and positively related to the amount of resource added. The fact that these relationships were not statistically different among the lakes suggests that leucine incorporation was driven by the added resources rather than other factors that could have affected the experiment. For

example, variations in lake pH could have impacted the amount of resources taken up in the bioassays (del Giorgio and Davis 2003; Li et al., 2012). Our blank bioassays further confirmed the dependency between leucine incorporation and limiting resource concentration by showing that virtually no leucine incorporation occurred when the limiting resource was lacking in the growth media.

We used the leucine incorporation method as proxy for bacterial growth and related it to bioavailable resource

concentrations based on the premise that this process measures the rate of bacterial protein synthesis (Kirchman et al., 1985). Because proteins are large macromolecules within bacterial cells (approximately half of bacterial dry weight), they represent a substantial fraction of the resource uptake and its consequent conversion into biomass. Also, to carry out protein synthesis, bacteria use both C and N; nitrogenous compounds are taken up from the growth media to build proteins with energy obtained from C substrates. Phosphorus is also used in the process as it is crucial for controlling the adenosine triphosphate-

adenosine diphosphate cycle, which provides energy for the intracellular molecular synthesis. Due to the critical role that these three elements play within protein synthesis, our results represent an unequivocal relationship between resource availability and the amount of protein synthetized. We measured resource bioavailability over a time period of seven days and the major part of the resource pool was exhausted within three days (Fig. 1). In the context of bioavailbility assessments, seven days is a relatively short period and repeated bacterial regeneration of resources was in this way avoided (Cho et al.,

1996). Although there may have been some resource recycling, our bioavailability estimates are automatically corrected for this artifact as these were calculated based on a leucine incorporation per unit of bioavailable resource relationship that was estimated for the exact same time period.





An important advantage of estimating nutrient bioavailability with our method is that uncertainties inherent to conversion factors (such as those used in bacterial production and flow cytometry) are avoided (Calvo-Diaz and Moran, 2009). This is because our integrated leucine amounts are directly transformed into bioavailable resource units through the slope of a specific load-growth relationship that is based on the growth of the exact same bacterial community exposed to a similar

media. Still, the design of our experiment could lead to possible sources of errors in estimates. For example, reference assays were performed at one occasion and used to interpret actual nutrient bioavailability at other occasions. This means that if BGE varied during the studied period it, could result in differences in the amounts of leucine incorporated. We dealt with this possible shortcoming by designing our bioassays such that resource use efficiency would be maximized (by strongly inducing resource limitation; Jansson et al., 2006) and thus, possible variations in resource use efficiency most likely did not

play a substantial role on rates of leucine uptake (Fig. 5). In addition, the fact that glucose was used as reference source of C and energy in the calibration could lead to an overestimation of the standard C growth curves and possibly result in conservative estimates of bioavailable C. For example, glucose additions could have supported the part of the community with the fastest growth and therefore results may not compare to results from a community that was instead exposed to a natural substrate. Nonetheless, when comparing the amount of leucine incorporated by our standard bacterial community per

unit of bioavailable glucose with amounts of leucine incorporated by different lake communities per unit of natural bioavailable substrate (Fig. 5), we show that, on average, our growth response was only slightly higher than the growth response in experiments based on alternative bioassay methods (Fig.5). Thus, our resource bioavailable estimates presented here are most likely conservative but realistic.

## 5 Conclusion

Ongoing changes in the global C, N and P cycles have the capacity to modify the chemical conditions and nutrient balance of receiving waters (Finzi et al., 2011). Yet the effects of these changes on basal productivity and food webs of many inland waters remain difficult to predict. We claim that to better forecast the impact of such changes, it is important that we refine how we consider and measure the stoichiometry of the main elements available to support aquatic production. This study contributes to our general understanding of resource dynamics in DOM-rich systems. Based on bioavailable resource ratios

determined with a single approach, we show that resource bioavailability increases from C to N and N to P. P availability in these systems may, thus, be likely considerable higher than previously thought. This finding particularly calls into question whether results from most enrichment experiments done so far, which often show that P additions stimulate BP, are applicable to DOM-rich systems (Jansson et al., 2001). In addition, our findings reinforce the idea that despite boreal waters being DOM-rich, C availability still represents the major constraint to BP in humic waters. This means that expected future

changes in the amount or character of C delivered to boreal surface waters will most likely drive changes in BP, which subsequently affects abiotic conditions, the biotic structure, and ecosystem functioning of freshwaters.





**Competing interests**

The authors have no conflict of interest to declare.

**Acknowledgements**

The authors thank Anders Jonsson, Juan Pablo Niño, Karla Münzer, Julia Jakobsson and Lina Allesson for providing help

with the sample analysis. We also thank Anne Deininger and Marcus Klaus for help with lake water sampling and shipping.
ARAS would like to acknowledge the ClimBEco Graduate School for funding a research visit. The Crafoord Foundation
(grant #20120626), KSLA (grant #H13-0020-GBN), FORMAS (grant #217-2010-126) and Helge Ax:son Johnson's
Foundation (grant #140622) contributed with funding of the study via MB. RG was supported by The Carl Trygger
Foundation for Scientific Research (grant #CTS12:147) and A.-K.B. by FORMAS (grant #215-2010-992).

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



**Table 1.** Descriptive lake data and concentrations of total dissolved nitrogen (TDN), nitrate ($NO_3^-$), ammonium ($NH_4^+$), total phosphorus (TP), phosphate ($PO_4$-P) and dissolved organic carbon (DOC) given as minimum and maximum values observed during the experimental period.

| Variables | Övre Björntjärnen | Lillsjöliden | Struptjärnen | Stortjärnen |
|---|---|---|---|---|
| Location (latitude [N], longitude [E]) | 64°7'23.53"N, 18°46'43.04"E | 63°50'41.71"N, 18°36'59.62"E | 64° 1'22.62"N, 19°29'21.18"E | 64°15'42.11"N, 19°45'44.73"E |
| Lake surface area (ha) | 4.8 | 0.8 | 3.1 | 3.9 |
| Maximal depth (m) | 9.5 | 5.2 | 5.8 | 6.7 |
| Total catchment area (ha) | 284 | 25 | 79 | 82 |
| Wetland coverage (%) | 16 | 2 | 4 | 12 |
| Forest coverage (%) | 84 | 98 | 96 | 88 |
| DOC (mg $L^{-1}$) | 18-29 | 13-19 | 19-25 | 19-27 |
| TDN (µg $L^{-1}$) | 376-502 | 336-501 | 360-521 | 355-598 |
| DIN (µg $L^{-1}$) | 5-35 | 10-40 | 3-43 | 4-35 |
| TP (µg $L^{-1}$) | 8-25 | 4-15 | 8-25 | 7-15 |
| $PO_4$-P (µg $L^{-1}$) | 1-8 | 0-4 | 0-3 | 0-2 |



**Table 2. (a) Bioavailable dissolved organic carbon, (b) bioavailable total nitrogen, and (c) bioavailable total phosphorus on seven sampling dates (columns). Values show means of five analytical replicates and standard deviations are provided within parentheses. Shared index letters within rows identify dates significantly different from each other (p < 0.05) which were determined by the Kruskal-Wallis h and Dunn's post-hoc test.**

| Lake | Sep 2012 | Oct 2012 | Jul 2013 | Jun 2014 | Jul 2014 | Aug 2014 | Sep 2014 |
|---|---|---|---|---|---|---|---|
| *a) BDOC, µg C L$^{-1}$* | | | | | | | |
| Övre Björntjärnen | 273$^a$ (143) | | 371$^{bcd}$ (64) | 420$^{aefg}$ (60) | 248$^{be}$ (39) | 243$^{cf}$ (22) | 216$^{dg}$ (21) |
| Lillsjöliden | 471 (435) | 552$^{ab}$ (338) | 334$^{cde}$ (49) | 176$^{ac}$ (36) | 205$^d$ (7) | 215 (17) | 176$^{be}$ (36) |
| Struptjärnen | 361$^a$ (327) | | 432$^b$ (93) | 692$^{acd}$ (85) | 337$^e$ (27) | 178$^c$ (21) | 107$^{bde}$ (6) |
| Stortjärnen | 319$^a$ (210) | 428$^b$ (228) | 283$^c$ (49) | 301$^d$ (35) | 213$^e$ (15) | 104$^{abcdf}$ (8) | 406$^{ef}$ (130) |
| *b) BDN, µg N L$^{-1}$* | | | | | | | |
| Övre Björntjärnen | 209$^{ab}$ (13) | | 74$^c$ (13) | 61$^a$ (6) | 84$^d$ (14) | 73$^e$ (5) | 23$^{bcde}$ (1) |
| Lillsjöliden | 287$^{abc}$ (10) | 232$^{def}$ (24) | 111$^{gh}$ (7) | 33$^{adg}$ (10) | 64$^{be}$ (5) | 89$^a$ (6) | 51$^{cfh}$ (3) |
| Struptjärnen | 259$^{abc}$ (6) | | 107$^{ad}$ (28) | 220$^e$ (14) | 273$^{dfg}$ 28 | 60$^{bf}$ (6) | 37$^{ceg}$ (2) |
| Stortjärnen | 188$^{ab}$ (15) | 206$^{cef}$ (18) | 82$^{ac}$ (6) | 67$^{be}$ (5) | 84$^f$ (5) | 28$^{acfg}$ (3) | 119$^g$ (38) |
| *c) BDP, µg P L$^{-1}$* | | | | | | | |
| Övre Björntjärnen | 9$^{abc}$ (1) | | 5$^{ad}$ (0) | 5$^{be}$ (0) | 9$^{def}$ (1) | 7$^g$ (0) | 3$^{cfg}$ (0) |
| Lillsjöliden | 3$^{ab}$ (0) | 3$^c$ (0) | 2$^{def}$ (0) | 2$^{cgh}$ (0) | 10$^{acd}$ (2) | 7$^{beg}$ (1) | 6$^{fh}$ (0) |
| Struptjärnen | 6$^{ab}$ (1) | | 6$^c$ (0) | 9$^{ad}$ (1) | 16$^{bce}$ (2) | 7$^e$ (1) | 4.$^{de}$ (0) |
| Stortjärnen | 0$^{ab}$ (0) | 0$^{cdef}$ (0) | 1$^{gh}$ (0) | 10$^{acg}$ (2) | 12$^{bdh}$ (2) | 6$^e$ (0) | 5$^f$ (2) |



**Table 3. Resource bioavailability in relation to the total resource pool, shown as percent bioavailable dissolved organic carbon (BDOC), bioavailable dissolved nitrogen (BDN) and bioavailable dissolved phosphorus (BDP). The data is divided into two groups which show average results for rivers with more than 10 mg C $L^{-1}$ (rivers$_{>10\ mg\ C\ L}^{-1}$; $n = 3$) and rivers with less than 10 mg C $L^{-1}$ (rivers$_{<10\ mg\ C\ L}^{-1}$; $n = 4$). Average element ratios of carbon to nitrogen (C : N), carbon to phosphorus (C : P), nitrogen to phosphorus (N : P) are calculated in molar for total (tot) and bioavailable resource fractions (bio). Ratios of dissolved inorganic nitrogen to phosphate (DIN : $PO_4$-P) are also provided. Standard deviations are given within parentheses.**

| Variable | rivers$_{>10\ mg\ C\ L}^{-1}$ | rivers$_{<10\ mg\ C\ L}^{-1}$ |
|---|---|---|
| BDOC (%) | 2 (1) | 3 (2) |
| BDN (%) | 48 (16) | 36 (20) |
| BDP (%) | 20 (12) | 31 (45) |
| C:N (bio) | 1 (1) | 2 (1) |
| C:N (total) | 26 (5) | 24 (13) |
| C:P (bio) | 319 (287) | 523 (795) |
| C:P (total) | 1722 (378) | 920 (93) |
| N:P (bio) | 294 (353) | 240 (251) |
| N:P (tot) | 70 (27) | 46 (21) |
| DIN: $PO_4$-P | 88 (68) | 2 (2) |





**Figure caption**

**Figure 1: Leucine incorporation rates over the incubation time for a blank incubation and five spikes of C (spike 1=330, spike 2=660, spike 3=1000, spike 4=1330 and spike 5=1500 µg C L$^{-1}$), N (spike 1=105, spike 2=133, spike 3=205, spike 4=305 and spike 5=405 µg N L$^{-1}$) and P (blank, spike 1=15.5, spike 2=18.8, spike 3=20.5, spike 4=30.5 and spike 5=40.5 µg P L$^{-1}$).**

**Figure 2: Measurements of leucine incorporation in relation to additions of bioavailable C (as $C_6H_{12}O_6$), N ($NH_4NO_3$) and P ($Na_2HPO_4$). Regression equations for all points pooled together: bioavailable C= 784x + 384 ($R^2 = 0.74$, $p < 0.0001$; $n = 20$); bioavailable N= 2667x + 159 ($R^2 = 0.75$, $p < 0.0001$, $n = 20$); bioavailable P = 67575x - 110 ($R^2=0.80$, $p < 0.0001$, $n = 20$). Note that each individual regression line in the figure has a better fit than the average regression line.**

**Figure 3: Proportion of organic non-bioavailable, organic bioavailable and inorganic nutrient shares of dissolved organic carbon (DOC), total dissolved nitrogen (TDN) and total phosphorus (TP) for all lakes and all sampling occasions (n=26).**

**Figure 4: Bioavailable (bio) and total (tot) ratios (in molar) of carbon to nitrogen (C:N) and carbon to phosphorus (C:P) for all**
15 **lakes and all sampling dates (n=26). Ratios of N:P are shown for total, bioavailable and inorganic (inorg) fractions. Different letters stand for significant differences (dependent t-test; $p < 0.05$; $n = 26$) among ratios.**

**Figure 5: Log-scale boxplots incubation show leucine amounts per unit of bioavailable nutrient measured with validation methods: bacterial respiration (C), cytometry (N) and harvesting of cells in filters (P). Diamonds are average values for the validation**
20 **methods and filled squares are average slope values for standard curves (same values as slopes in Fig. 2).**



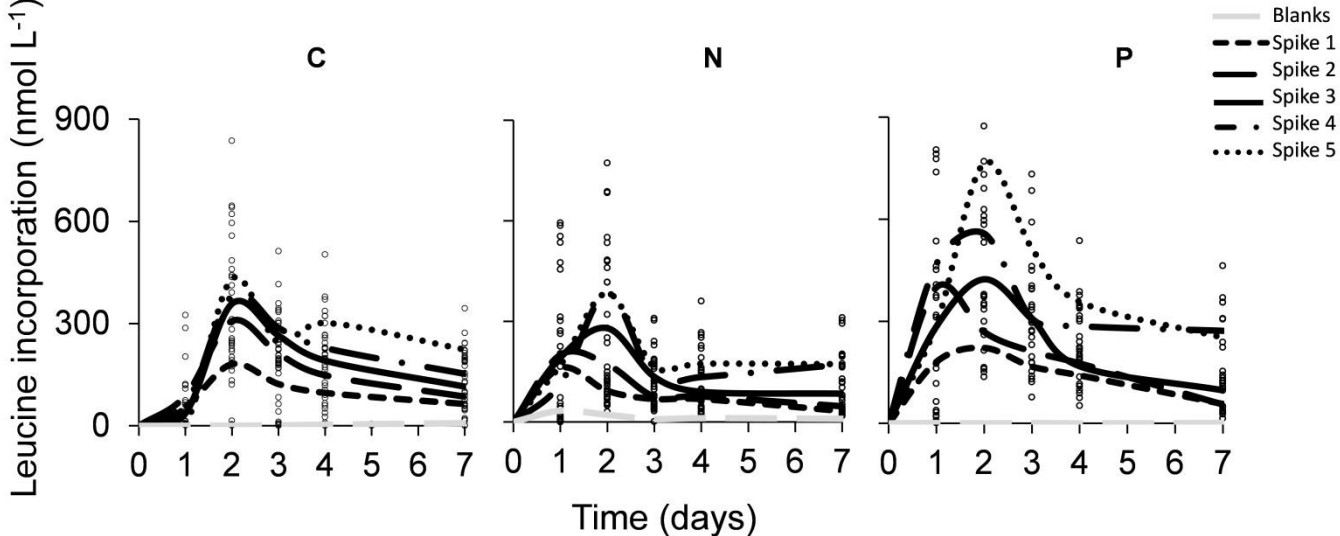

**Figure 1.**



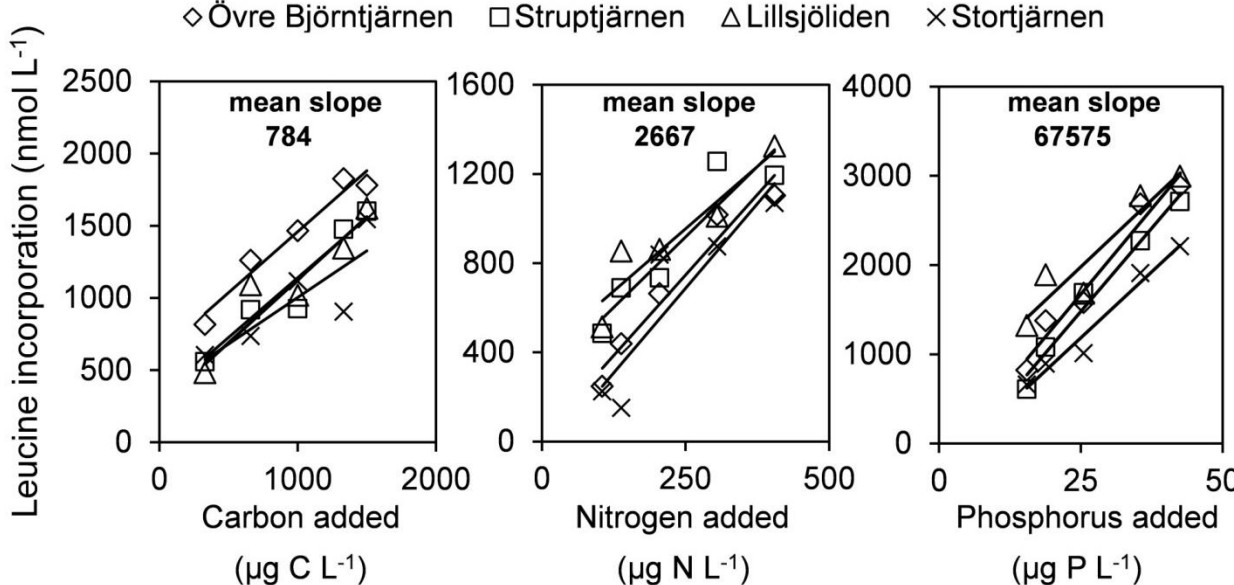

**Figure 2.**




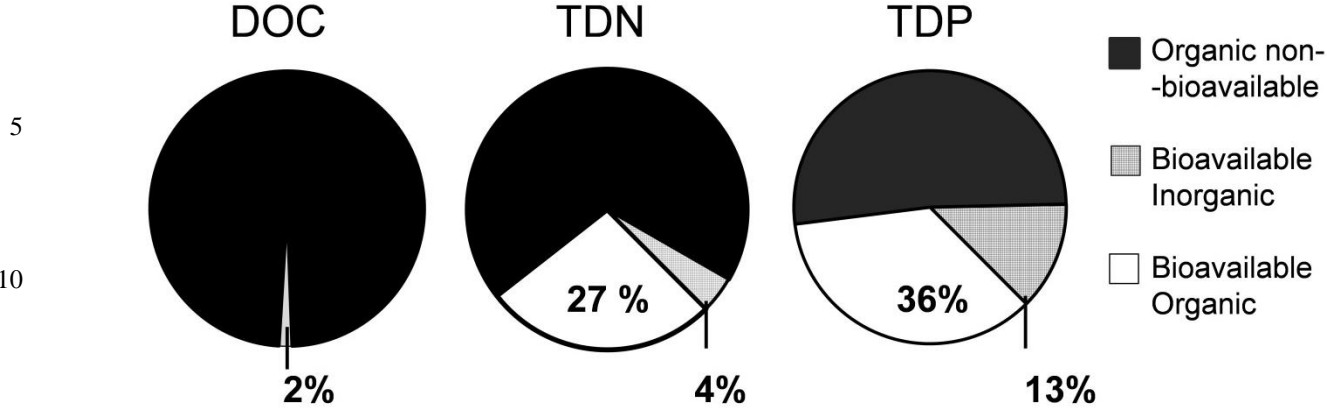

**Figure 3**.



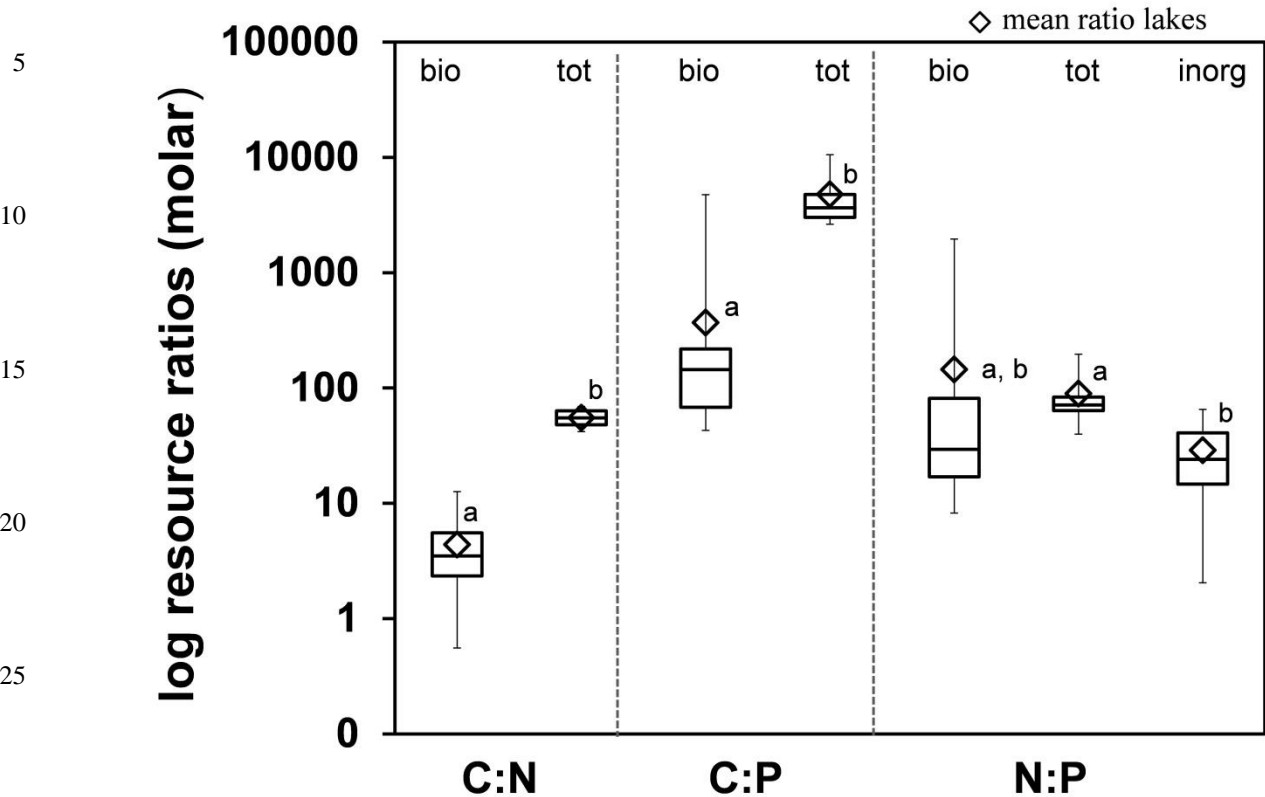

**Figure 4**.



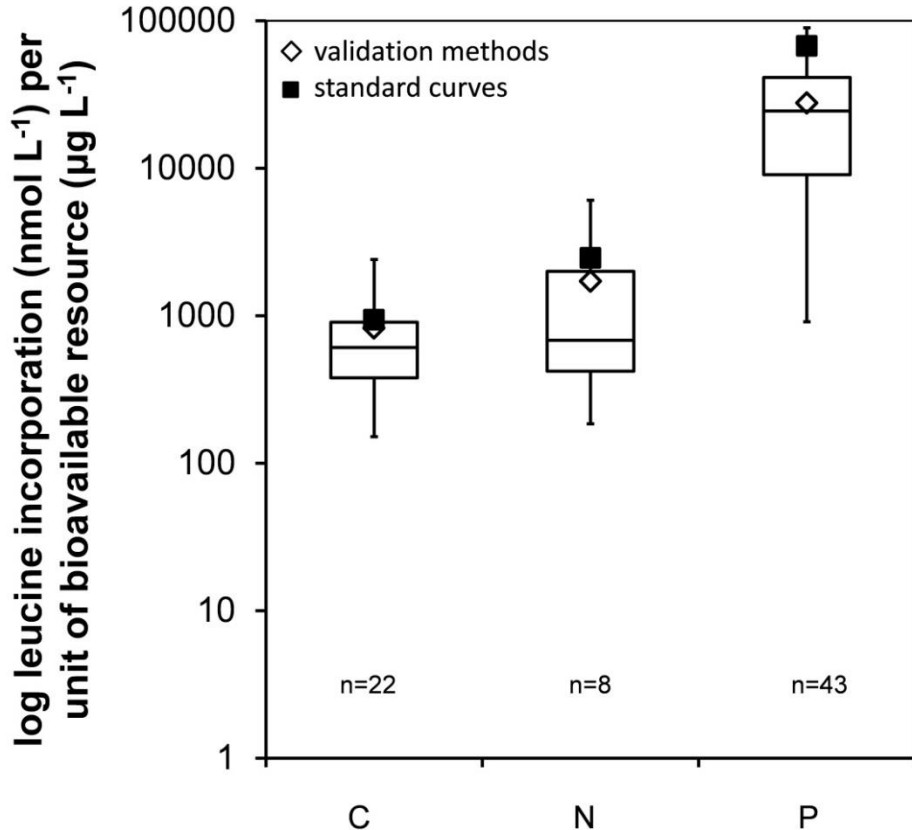

**Figure 5.**