# Peer review of "New insights on resource stoichiometry: assessing availability of carbon, nitrogen and phosphorus to bacterioplankton"

_Biogeosciences, 2016_

## Referee Comment (RC1) · Anonymous Referee #1 · 13 Dec 2016

The manuscript by Soares and others is a novel and important contribution to this topic. In particular, their innovative experimental approach offers an answer to the question: what resource stoichiometry to bacteria actually experience in situ, given that not all measurable forms are bio-available? The work was thoughtfully designed and executed and will be of interest to the readership of Biogeosciences.

Two areas require attention from the authors. First, the conclusion that C is limiting is not adequately supported by the manuscript in its present form (see below). Second, the uncertainties in bioavailable concentrations must be made more clear. Aside from these two areas, the paper is strong and the other comments are minor/clarification.

Page 1 Line 24. What is the evidence for this in the present study? Although the

resource stoichiometry derived from their results suggests that C will likely be limiting before N or P, this does not automatically mean that C is limiting. That extension of resource stoichiometry is applicable only if 1) the bacteria are resource-limited and not under top-down control; 2) the only potentially limiting resources are C, N, or P; and 3) the system is presumed to be at steady state resembling a chemostat.

Page 3 Line 8. While the long incubations have their shortcomings, it is overstated and confusing to say that these are not 'ecologically relevant timescales'. Certainly the majority of the consumption and respiration in fresh DOM happen in a matter of hours to days. However, longer-term degradation rates of more recalcitrant forms are of key importance. Specific to this study, the rapid rates of consumption observed are due to the high concentrations of CNP added and thus, the timescale of the experiment is not ecologically relevant. I suggest that the authors focus this section and justification on the multi-element aspect of their design, which is the important and novel part.

Page 3 Line 30. The third question seems certain to be true, and thus not informative as a question or hypothesis. Yet, quantifying this mismatch is important, so I suggest that the authors rewrite these questions.

Page 4 Line 10. By sampling the rivers at their outlet, much of the bioavailable forms have presumably been consumed in transit. What is the rationale for sampling far downstream from the sources of DOM?

Page 5 Line 2. This standardized inoculum has important implications for interpreting the results. Elaborate on why this single community was used as opposed to the communities present in the source water.

Page 5 Line 15-30. This experimental approach is rather involved. If space allows, the authors should include a schematic diagram that shows how they forced limitation by CNP and measured the response to addition of the limiting resource. Presumably this method is based on the Wright-Hobbie technique and thus it is important to show how the estimates of ambient concentrations were derived.

Page 5 Line 30. This approach requires high confidence in the regressions used. Uncertainty in the slope and intercept for the standard curves can be propagated to estimate the uncertainty in the ambient concentrations that are estimated. The authors should include such calculations of uncertainty for at least a few representative samples (perhaps as a supplement if space is limiting).

Page 5 Line 30. "The total amount of bioavailable nutrient taken up" is not precise. Especially for C, the nutrient need not be assimilated in order for the bacteria to exhibit a growth response.

Page 6 Line 15. The use of complementary validation methods is an important strength of this paper. Well done.

Page 6 Line 32. This method of calculating cellular N content is strange. What are the assumptions of this method? At the least it assumes that all of the added N is assimilated and that no other N is used.

Page 7 Line 5. The validation method used for P availability is more straightforward than for N. Why not use this method for N also? Additionally, were these fitler-P measurements corrected/checked for phosphate binding to the filter?

Page 7 Line 30. Needs clarification. No difference between slopes for C, N, and P or among lakes? Also, it is unclear why the regressions were performed individually for each analytical replicate instead of using all of the analytical replicates for a given site/date. From what I can tell, the standard curves were computed individually for each of five analytical replicates and then the standard deviation of their estimates is presented in table 2?

Page 9 Line 20. Were the total and bioavailable concentrations (or elemental ratios) positively correlated?

Page 9 Line 23. Again, what is the evidence that C was most limiting, or even limiting at all? The traditional lines of evidence for this (single nutrient bioassays) are not

presented, so this is either inferred from the stoichiometry estimated for resources or from the low proportional bioavailability of C compared to N and P. Neither of these shows that C was the strongest limiting factor. Pease elaborate on this and explain 1) the assumptions used for this claim and 2) the specific evidence from this study

Page 10 Line 33. There are many other factors related to seasonality that could eplain this (light, plant production, hydrology, etc), so how can you conclude that soil microbial activity is the predominant driver? Overall, I found this discussion of seasonality too speculative

Page 11 Line 27. These calculations seem to be the core of the argument that C is limiting and thus require elaboration. Even then, this only shows that C is more likely to be limiting than N or P, but does not show that C was in fact limiting at ambient concentrations.

Moreover, the ranges here are so large that they are not really meaningful. Why not use the ratio of slopes presented in figure 2 to estimate the relative consumption rates of CNP? In your calculations, you already assume that the ratio of leucine:cell is invariant, so the ratio of 1/C-slope to 1/P-slope (=86) is the ratio of C consumption to P consumption when those elements are limitng. No?

In both the lakes and the rivers, the DOM pools have already undergone much degradation by bacteria, light, and reactive oxygen. This needs to be acknowledged, or better yet, discussed in some detail.

Page 13, line 1. Avoiding these uncertainties is important, but those are typically on the order of a few percent and can be constrained by experimental validation. Without a robust analysis of the resulting uncertainties from the present approach, it is not possible to discern which method is advantageous. Form Table 2 and Figure 1/2, it appears that the uncertainty in concentration estimated for a single date/site is large. Without such an analysis of the uncertainty in the final estimates, I suggest that the authors focus on the multi-element aspects of their study

Figure 4. What do the diamonds represent in this figure?

Figure 5. The vertical axis scale should be fitted to the range of data presented.
* * *

---

## Referee Comment (RC2) · Anonymous Referee #2 · 23 Dec 2016

General Comments

This manuscript presents the results from a test of a new method of determining the relative bioavailability of carbon, nitrogen and phosphorus for lake and riverine bacterioplankton. The technique, which combines radiolabeled leucine incubations with reciprocal nutrient amendments, is a novel approach to backing out the proportion of total dissolved C, N and P that bacteria can rapidly take up if other factors are not limiting. The authors test the approach with seasonal samples from four Swedish lakes and single-date samples from seven rivers. Overall, the authors provide a very interesting analysis and the paper is in good shape. Please see below for my specific and technical comments. The only general comment that I would make is that the approach

explicitly considers bioavailability in the absence of any co-limitation. In other words, the method cannot incorporate any interactions between limiting factors. While this may be a necessary shortfall of the approach, its significance perhaps deserves some thought and maybe some treatment in the discussion.

Specific Comments

1) Page 1, line 17-18: Make sure the readers know that these percentages are based on the initial concentrations. I know that this might sound obvious, but I was initially confused about whether these were percentages of final (post-incubation) or initial (pre-incubation) amounts.

2) Page 5, line 2: Where exactly was the inoculum sampled? And how could it have been sampled only once, given that the lake and river samples were collected over a lengthy period and the incubations run soon after each sample collection? Was it maintained in the laboratory?

3) Page 5, line 8: Could there be an effect of incubating bacterioplankton in such a small volume of water? Could biofilms on the walls of these small vials start to have a disproportionate impact on the results?

4) Page 5, line 16: Maybe I'm missing something, but why didn't the controls consist of lake water without any added C, N or P?

5) Page 5, line 24: Presumably these standard curves would be system-specific? Or at least limited to similar environments within a region? Some discussion of should perhaps be added to the discussion.

6) Page 6, line 24 to page 7, line 8: It sounds like these methods assume no changes in cellular stoichiometry with nutrient availability (i.e. elemental homeostasis).

7) Page 11, line 5: Is this consistent with turnover rates of these elements in these ecosystems?

8) Page 11, line 23: Perhaps mention threshold elemental ratios here, as well as the work that has focused on them in bacteria (Sinsabaugh, Chrzanowski, etc).

Technical Corrections

1) Page 1, line 14: "...purpose of exhausting the pools..."

2) Page 1, line 16: "base-flow"

3) Page 2, line 20: Delete one of these extraneous uses of "on"

4) Page 2, line 26: Why "re-growth"? Wouldn't it be simpler just to call these "growth assays"?

5) Page 3, line 11: Delete "single"

6) Page 3, line 18: "...purpose of rapidly exhausting the pools..."

7) Page 3, line 23: "N-starved"

8) Page 3, line 28: Replace "shares" with "proportions"

9) Page 3, line 34: Replace "compromises" with "comprises"

10) Page 4, line 16: It says "runoff" here, but the units in the next line suggest that the authors mean discharge.

11) Page 4, line 30: "climate-controlled"

12) Page 4, line 32: "1000-ml"

13) Page 5, line 2: There's an extra "the" in this sentence. It's also not very clear (see my comment in Specific Comments above).

14) Page 5, line 6: "N-limiting"

15) Page 5, line 7: "P-limiting"

16) Page 5, line 19: "C-limiting"

17) Page 6, line 10: "seven-day"

18) Page 6, line 20: "sensor spots"

19) Page 6, line 25: "N-starved"

20) Page 7, line 1: "N-amended"

21) Page 7, line 23: "pools"

22) Page 8, lines 1 and 7: "tests"

23) Page 9, line 6: "...were validated..."

24) Page 9, line 16: Replace "neither" with "nor"

25) Page 9, line 19: "...as a driver of..."

26) Page 10, line 10: Delete "also"

27) Page 10, line 12: "single-element"

28) Page 12, line 6: Replace "Whereas" with "However"

29) Page 12, line 7: "...similar to the lake..."

30) Page 12, line 19: "...method as a proxy..."

31) Page 12, line 23: Replace "media" with "medium"

32) Page 12, line 27: "synthesized"

33) Page 12, line 28: "bioavailability"

34) Page 13, line 5: Replace "media" with "medium"

35) Page 13, line 7: Delete comma

36) Fig. 1: Why are the data points from the different treatments not differentiated here?

---

## Author Comment (AC1) · 3 Feb 2017

Response to Referee 1

We thank Referee 1 for constructive and relevant comments to the manuscript and for helping us to improve it. We addressed all comments below.

General comment Referee 1:" The manuscript by Soares and others is a novel and important contribution to this topic. In particular, their innovative experimental approach offers an answer to the question: what resource stoichiometry to bacteria actually experience in situ, given that not all measurable forms are bio-available? The work was thoughtfully designed and executed and will be of interest to the readership of Biogeosciences.
Two areas require attention from the authors. First, the conclusion that C is limiting is not adequately supported by the manuscript in its present form (see below). Second, the uncertainties in bioavailable concentrations must be made more clear. Aside from these two areas, the paper is strong and the other comments are minor/clarification."

Referee comment 1:" Page 1 Line 24. What is the evidence for this in the present study? Although the resource stoichiometry derived from their results suggests that C will likely be limiting before N or P, this does not automatically mean that C is limiting. That extension of resource stoichiometry is applicable only if 1) the bacteria are resource-limited and not under top-down control; 2) the only potentially limiting resources are C, N, or P; and 3) the system is presumed to be at steady state resembling a chemostat."

**Authors' comment: We agree with the Referee and acknowledge that we do not present direct evidence showing C limitation. We have therefore reformulated all sentences in this regard, clarifying that C was the least bioavailable element (in % bioavailability) out of the three key macronutrients that we work with. We also have now made clear that our bioavailability estimates are informative of maximum potential bioavailability under specific conditions, i.e. when all other macro- and micronutrients of relevance are in excess. Thus, while we can state that access to bioavailable C in our samples tended to be scarce relative to the microbial need and access to N and P, the apparent C limitation is not directly transferrable to natural systems, especially not when considering the dynamic nature of natural ecosystems and the potential presence of top-down controls and/or micronutrient limitation.**

Referee comment 2: Page 3 Line 8. While the long incubations have their shortcomings, it is overstated and confusing to say that these are not 'ecologically relevant timescales'. Certainly the majority of the consumption and respiration in fresh DOM happen in a matter of hours to days. However, longer-term degradation rates of more recalcitrant forms are of key importance. Specific to this study, the rapid rates of consumption observed are due to the high concentrations of CNP added and thus, the timescale of the experiment is not ecologically relevant. I suggest that the authors focus this section and justification on the multi-element aspect of their design, which is the important and novel part.

**Authors' comment: We agree with the Referee on the ecological importance of long-term degradation of more recalcitrant DOM, particularly in systems with long water residence times. However, resource bioavailability measured with long-term incubations does not reflect readily bioavailable pool sizes that control bacterial metabolism at any given moment. Moreover, during long incubation periods various factors can interfere with the uptake of bioavailable resources. For example, the dynamics of viruses and the development of toxic conditions that can appear from repeated bacterial regeneration of resources can interfere in long-term measurements (Cho et al., 1996). By using our seven-day approach and by maximizing bacterial metabolism, we reduce the incubation length to a minimum and sufficient time period during which bacteria take up most of the readily bioavailable pool (Fig. 1). Our estimates can be used to understand the potential C, N and P bioavailability, as they are performed during "ecologically relevant timescales" in this regard. In our revised manuscript we clarify that the relevance refers to how meaningful the measurements are for understanding the direct controls of bioavailable nutrient pools on short-term metabolism– not the controls the nutrient pools may have months ahead in time.**

Referee comment 3: Page 3 Line 30. The third question seems certain to be true, and thus not informative as a question or hypothesis. Yet, quantifying this mismatch is important, so I suggest that the authors rewrite these questions.

**Authors' comment: The third question was changed to "By how much do total C:N, C:P and N:P ratios exceed bioavailable C:N, C:P and N:P ratios".**

Referee comment 4: Page 4 Line 10. By sampling the rivers at their outlet, much of the bioavailable forms have presumably been consumed in transit. What is the rationale for sampling far downstream from the sources of DOM?

**Authors' comment: Our goal was to capture bioavailability patterns across a landscape gradient with different boreal freshwater properties (see first manuscript version page 3 lines 31-32) and not to determine the amount of bioavailable element coming from terrestrial soils. Nonetheless, in the revised manuscript we will clarify that Swedish river systems generally represent substantial water renewal along the watercourse pathways from the Scandes in the west toward the Baltic Sea in the East. In fact it has been suggested that water renewal in running waters offsets the loss of DOC in Swedish**

**lakes such that rivers in Sweden generally does not have much older DOC than lakes (Muller et al., 2013).**

Referee comment 5: Page 5 Line 2. This standardized inoculum has important implications for interpreting the results. Elaborate on why this single community was used as opposed to the communities present in the source water.

Authors' comment: **We wanted to ensure that differences in bacterial community composition did not influence our estimates of resource bioavailability (Martinez et al., 1996). This was achieved by using a standard bacterial community in all our assays. We have now explicitly motivated the use of a single bacterial assemblage as inoculum in the manuscript. By using a pooled inoculum representing both headwater inlet and lake water from four different lakes with different properties, we ensured a high diversity of the microbes used to inoculate.**

Referee comment 6: "Page 5 Line 15-30. This experimental approach is rather involved. If space allows, the authors should include a schematic diagram that shows how they forced limitation by CNP and measured the response to addition of the limiting resource. Presumably this method is based on the Wright-Hobbie technique and thus it is important to show how the estimates of ambient concentrations were derived."

**Authors' comment: We agree that it is important to include a schematic diagram to help to better visualize our approach. We have added a schematic diagram of the method to the supplementary material in the revised manuscript.**

Referee comment 7: "Page 5 Line 30. "The total amount of bioavailable nutrient taken up" is not precise. Especially for C, the nutrient need not be assimilated in order for the bacteria to exhibit a growth response."

**Authors' comment: It is true that the leucine incorporation itself reflects growth – not respiration. However, the way our method is designed, the total leucine incorporation is recalculated into absolute units of bioavailable carbon with help from the standard curves. This works as we have the same consistent slope of the standard curves in all lakes and across all C spike levels, implying that our experiment creates conditions with fixed (maximized) growth efficiency. Thus, we think that our sentence is well formulated. We used leucine incorporation as an experimental response variable of all bioavailable element uptake, which in the case of C can be used either for growth or respiration.**

Referee comment 8: "Page 6 Line 15. The use of complementary validation methods is an important strength of this paper. Well done."
**Author's comment: Thank you for pointing this out.**

Referee comment 9: "Page 6 Line 32. This method of calculating cellular N content is strange. What are the assumptions of this method? At the least it assumes that all of the added N is assimilated and that no other N is used."

**Authors' comment: This method encompasses several assumptions: 1) bacterial growth in the bioassays was effectively limited by N, 2) different N compounds yield similar bacterial biomass increases, 3) all bioavailable N was assimilated when bacterial growth ceased and 4) N bioavailability was independent from the bacterial inocula. The paper from Stepanauskas et al. (1999) describes in detail the experimental setup and the method's assumptions.**

Referee comment 10: "Page 7 Line 5. The validation method used for P availability is more straightforward than for N. Why not use this method for N also? Additionally, were these fitler-P measurements corrected/checked for phosphate binding to the filter?"

**Authors' comment: It is not possible with any standard instrumentation to directly measure changes in absolute concentrations of bioavailable N (and C) with the same analytical precision as routine methods used to determine P (molybdenum blue method, microgram accuracy). In the revised manuscript we add a short section describing limitations and possibilities offered by different bioavailability determination methods based on Berggren et al. (2015).**

**Estimates of P bioavailability were corrected for potential P filter content, binding of dissolved P species, and abiotic formation of particles (Jansson et al., 2012).**

Referee comment 11: "Page 7 Line 30. Needs clarification. No difference between slopes for C, N, and P or among lakes? Also, it is unclear why the regressions were performed individually for each analytical replicate instead of using all of the analytical replicates for a given site/date. From what I can tell, the standard curves were computed individually for each of five analytical replicates and then the standard deviation of their estimates is presented in table 2?"

**Authors' comment: We have changed the sentence on page 7 line 30 from the previous manuscript to: "Since there were no statistical differences between the slopes among lakes for each resource (ANOVA, p > 0.44, n = 20), slopes were averaged for each resource nutrient across lakes."**

**We first performed the regressions individually (Figure 2), because we wanted to test whether there were differences in the bacterial response to nutrient additions between the different lakes. Since we found no statistically significant differences between lake slopes (this is mentioned on page 7 line 29 and page 12 lines 13-14), we combined all data points and performed a new regression for each element based the entire dataset. This rendered the "mean slope" given on Figure 2 (C slope=784 nmol $L^{-1}$ per μg C $L^{-1}$, N=slope 2667 μg N $L^{-1}$, P slope=67575 μg P $L^{-1}$).**

**In table 2, the mean slope of the standard curves was used to translate amounts five replicate measurements of leucine uptake. The standard deviation of the estimates is given within brackets.**

Referee comment 12: "Page 9 Line 20. Were the total and bioavailable concentrations (or elemental ratios) positively correlated?"

**Authors' comment: No, there was no correlation between the total and bioavailable concentrations. This will be clarified in the revision.**

Referee comment 13: "Page 9 Line 23. Again, what is the evidence that C was most limiting, or even limiting at all? The traditional lines of evidence for this (single nutrient bioassays) are not presented, so this is either inferred from the stoichiometry estimated for resources or from the low proportional bioavailability of C compared to N and P. Neither of these shows that C was the strongest limiting factor. Pease elaborate on this and explain 1) the assumptions used for this claim and 2) the specific evidence from this study"

**Authors' comment: We agree with the Referee that we do not have the evidence needed to claim that C is limiting in boreal waters (see answer to Referee comment 1). We have changed the sentence from page 9 line 21 from the previous manuscript version "Surprisingly, in these systems where absolute surface water DOC concentrations are large, C bioavailability was low and was the strongest limiting factor for heterotrophic aquatic production." to "In these systems where absolute surface water DOC concentrations are large, relative C bioavailability (%) was the lowest, relative to that of N and P."**

Referee comment 14: Page 10 Line 33. There are many other factors related to seasonality that could explain this (light, plant production, hydrology, etc), so how can you conclude that soil microbial activity is the predominant driver? Overall, I found this discussion of seasonality too speculative

**Authors' comment: We agree that important role of other seasonal factors for the amount of bioavailable dissolved organic carbon measured in our study. We have now removed the sentences from the previous manuscript page 10 line 30 to page 11 line 1.**

Referee comment 15: Page 11 Line 27. These calculations seem to be the core of the argument that C is limiting and thus require elaboration. Even then, this only shows that C is more likely to be limiting than N or P, but does not show that C was in fact limiting at ambient concentrations.

Moreover, the ranges here are so large that they are not really meaningful. Why not

use the ratio of slopes presented in figure 2 to estimate the relative consumption rates of CNP? In your calculations, you already assume that the ratio of leucine:cell is invariant, so the ratio of 1/C-slope to 1/P-slope (=86) is the ratio of C consumption to P consumption when those elements are limiting. No?

In both the lakes and the rivers, the DOM pools have already undergone much degradation by bacteria, light, and reactive oxygen. This needs to be acknowledged, or better yet, discussed in some detail.

**Authors' comment: We agree with the Referee. We have thus, reformulated our conclusion and all statements related to C limitation (please see also answers to Referee comment 1 and 13).**

**We decided to exclude the calculations in question from the manuscript. The reviewer's suggestion of assuming a ratio for the C consumption in relation to P consumption base on Figure 2 is interesting, but it would not work to translate the uptake ratios of the experiment to nature as our methodological approach based on inducing strong limitation of each element is boosting the nutrient use efficiency to max (Jansson et al., 2008). In nature, the nutrient use efficiency (including C use efficiency = BGE) is highly variable and probably much lower than in our experiment. We therefore rather prefer to remove the discussion of C limitation of BP.**

**We partly agree with the Referee regarding the loss of most of the riverine bioavailable pool, but see our reply to point 4 above. However as we targeted the medium-term bioavailable resource pool, we do not consider this to be a problem. Nonetheless, the discussion of our revised manuscript will expand on the subject of how the river system differ in nature compared to the lakes, and what the implications of that are in the context of macronutrient biovailability.**

Referee comment 16: "Page 13, line 1. Avoiding these uncertainties is important, but those are typically on the order of a few percent and can be constrained by experimental validation. Without a robust analysis of the resulting uncertainties from the present approach, it is not possible to discern which method is advantageous. Form Table 2 and Figure 1/2, it appears that the uncertainty in concentration estimated for a single date/site is large. Without such an analysis of the uncertainty in the final estimates, I suggest that the authors focus on the multi-element aspects of their study"

**Authors' comment: As suggested we will focus our discussion on the multi-element aspect of our study. Thus, we have removed from the previous manuscript lines 1 to 5 page 13.**
Referee comment 17: "Figure 4. What do the diamonds represent in this figure?"
**Authors' comment: The diamonds represent average resource ratio values for the lakes for all dates (n=26). We have added to Figure's 4 caption the following sentence: "Data shown as boxplots and includes mean as diamonds.**

Referee comment 18: "Figure 5. The vertical axis scale should be fitted to the range of data presented."

**Authors' comment: Vertical axis scale has been changed from 1 to 100000 to 100 to 100000.**

**References:**

Bushaw, K. L., Zepp, R. G., Tarr, M. A., SchulzJander, D., Bourbonniere, R. A., Hodson, R. E., Miller, W. L., Bronk, D. A., and Moran, M. A.: Photochemical release of biologically available nitrogen from aquatic dissolved organic matter, Nature, 381, 404-407, 1996.

Berggren, M., Sponseller, R. A., Soares, A. R. A., and Bergstrom, A. K.: Toward an ecologically meaningful view of resource stoichiometry in DOM-dominated aquatic systems, Journal of Plankton Research, 37, 489-499, 10.1093/plankt/fbv018, 2015.

Cho, B. C., Park, M. G., Shim, J. H., and Azam, F.: Significance of bacteria in urea dynamics in coastal surface waters, Mar Ecol Prog Ser, 142, 19-26, 10.3354/meps142019, 1996.

Creed, I. F., McKnight, D. M., Pellerin, B. A., Green, M. B., Bergamaschi, B. A., Aiken, G. R., Burns, D. A., Findlay, S. E. G., Shanley, J. B., Striegl, R. G., Aulenbach, B. T., Clow, D. W., Laudon, H., McGlynn, B. L., McGuire, K. J., Smith, R. A., and Stackpoole, S. M.: The river as a chemostat: fresh perspectives on dissolved organic matter flowing down the river continuum, Canadian Journal of Fisheries and Aquatic Sciences, 72, 1272-1285, 10.1139/cjfas-2014-0400, 2015.

Gao, H. Z., and Zepp, R. G.: Factors influencing photoreactions of dissolved organic matter in a coastal river of the southeastern United States, Environmental Science & Technology, 32, 2940-2946, 10.1021/es9803660, 1998.

Jansson, M., Berggren, M., Laudon, H., and Jonsson, A.: Bioavailable phosphorus in humic headwater streams in boreal Sweden, Limnology and Oceanography, 57, 1161-1170, 2012.

Jansson, M., Hickler, T., Jonsson, A., and Karlsson, J.: Links between terrestrial primary production and bacterial production and respiration in lakes in a climate gradient in subarctic Sweden, Ecosystems, 11, 367-376, 2008.

Jansson, M., Berggren, M., Laudon, H., and Jonsson, A.: Bioavailable phosphorus in humic headwater streams in boreal Sweden, Limnology and Oceanography, 57, 1161-1170, 2012.

Martinez, J., Smith, D. C., Steward, G. F., and Azam, F.: Variability in ectohydrolytic enzyme activities of pelagic marine bacteria and its significance for substrate processing in the sea, Aquatic Microbial Ecology, 10, 223-230, 10.3354/ame010223, 1996.

Muller, R. A., Futter, M. N., Sobek, S., Nisell, J., Bishop, K., and Weyhenmeyer, G. A.: Water renewal along the aquatic continuum offsets cumulative retention by lakes: implications for the character of organic carbon in boreal lakes, Aquatic Sciences, 75, 535-545, 10.1007/s00027-013-0298-3, 2013.

Stepanauskas, R., Leonardson, L., and Tranvik, L. J.: Bioavailability of wetland-derived DON to freshwater and marine bacterioplankton, Limnology and Oceanography, 44, 1477-1485, 1999.

---

## Author Comment (AC2) · 3 Feb 2017

We thank referee 2 for constructive and relevant comments and suggestions of technical corrections, which helped us to improve the manuscript. Please find our response below.

General Comments

This manuscript presents the results from a test of a new method of determining the relative bioavailability of carbon, nitrogen and phosphorus for lake and riverine bacterioplankton.The technique, which combines radiolabeled leucine incubations with reciprocal nutrient amendments, is a novel approach to backing out the proportion of total dissolved C, N and P that bacteria can rapidly take up if other factors are not limiting. The authors test the approach with seasonal samples from four Swedish lakes and single-date samples from seven rivers. Overall, the authors provide a very interesting analysis and the paper is in good shape. Please see below for my specific and technical comments. The only general comment that I would make is that the approach explicitly considers bioavailability in the absence of any co-limitation. In other words, the method cannot incorporate any interactions between limiting factors. While this may be a necessary shortfall of the approach, its significance perhaps deserves some thought and maybe some treatment in the discussion.

**Authors' comment to the general comment: The Referee is correct. Our method determines the maximum pool sizes of readily bioavailable macronutrient fractions that can be used given that all other nutrients are provided in access. In the revised manuscript version we clarify that our bioavailability estimates are defined under these specific operational conditions. We also explain that, in order to translate the implications of the results to natural systems, factors like nutrient co-limitation and potential limitation by micronutrients or even top-down controls (e.g., grazing as pointed out by Reviewer 1) need to be taken into account.**

Specific Comments

Referee comment: 1) "Page 1, line 17-18: Make sure the readers know that these percentages are based on the initial concentrations. I know that this might sound obvious, but I was initially confused about whether these were percentages of final (post-incubation) or initial (pre-incubation) amounts."

**Authors' comment: We thank the Reviewer for pointing this out. This has been clarified in the new manuscript version.**

Referee comment: 2) "Page 5, line 2: Where exactly was the inoculum sampled? And how could it have been sampled only once, given that the lake and river samples were collected over a lengthy period and the incubations run soon after each sample collection? Was it maintained in the laboratory?"

**Authors' comment: The inoculum consisted of a mixture of water from both the epilimnion and inlet of the lakes. By including communities from several different sampling sites, we ensured a large microbial diversity on the inoculum. The inoculum was maintained in the fridge at approximately 4 ˚C. Because our experiment strongly maximized bacterial metabolism (selecting for fast-growing opportunistic bacteria), we do not think that the inoculum played a large role on the outcome of our experiment. Previous studies have further demonstrated bacterial growth to be independent of bacteria inocula (Tranvik and Hofle, 1987).**

Referee comment: 3) Page 5, line 8: Could there be an effect of incubating bacterioplankton in such a small volume of water? Could biofilms on the walls of these small vials start to have a disproportionate impact on the results?

**Authors' comment: We did not test or control for the potential development of biofilms in the tubes walls. However, based on the results for phosphorus presented in Figure 5, we can compare our measurement of the amount of leucine incorporation (normalized per unit of bioavailable P; filled square) with corresponding data extracted from Jansson et al. (2012; the box plot). In the latter case, Jansson et al. did not involve incubations in Eppendorf tubes but in much larger (700+ ml ) volumes. There was an overlap in magnitude of leucine incorporation when comparing these two data sources, but it can be noted that our measurements are in the upper range compared to those from Jansson et al. (2012). Biofilm accumulation could have potentially contributed to this difference in our incubation tubes. However, when looking at the time series of our incubations (Figure 1), it is clear that most of the leucine incorporation in our case happened already within 3 days, which should be a time-frame too short for substantial biofilm formation. Thus, we do not consider that biofilms strongly influenced our results.**

Referee comment: 4) Page 5, line 16: Maybe I'm missing something, but why didn't the controls consist of lake water without any added C, N or P?

**Authors' comment: Since our design is based on the idea of inducing strong limitation of the nutrient to be evaluated for maximum potential bioavailability, we did not consider relevant to incubate lake water without any nutrient additions. On the lines that the Reviewer refers to, we tested whether the inoculum or L16 added any bioavailable C, N and P to our assays. By using Mili-Q water instead of lake water, we made sure the inoculum and L16 were the only possible sources of limiting resource in our bioassays. At the same time, this also tested that leucine incorporation (or bacterial growth) was in fact controlled by the induced limiting resource and that no bacterial growth occurred in the absence of the bioavailable limiting resource (see previous manuscript version page 12 lines 16-18).**

Referee comment: 5) Page 5, line 24: Presumably these standard curves would be system-specific? Or at least limited to similar environments within a region? Some discussion of should perhaps be added to the discussion.

**Authors' comment: We did not find significant differences among standard curves for the different lakes (page 12 line 13 previous manuscript version), which is interesting since the lakes represent gradients in DOC and catchment features representative to a range of boreal conditions. Possibly, corresponding standard curves from the rivers could have been different from those in the lakes, but it would have been a major time-consuming effort to determine those curves for all of the rivers. Therefore, results from the rivers should be interpreted with caution, even if the rivers do not represent fundamentally different chemical conditions. We added a short section on the subject to the discussion part in the new manuscript.**

Referee comment: 6) Page 6, line 24 to page 7, line 8: It sounds like these methods assume no changes in cellular stoichiometry with nutrient availability (i.e. elemental homeostasis).

**Authors' comment: Yes the Reviewer is correct; an invariant cellular stoichiometry was assumed in the validation method used to calculate N bioavailability. This, as well as other method assumptions, has been scrutinized in Stepanauskas et al. (1999).**

**However, there were no assumptions regarding cellular stoichiometry for the method used to calculate P bioavailability since this was based on direct measurements of the content of P in bacterial growth cultures harvested from filters.**

Referee comment: "7) Page 11, line 5: Is this consistent with turnover rates of these elements in these ecosystems?"

**Authors' comment: We do not know of studies looking at the turnover rates of these elements in the soils in the study area. We have deleted the sentence in question and replaced it with a clearer sentence that brings into attention the main mechanism suggested by Jansson et al. (2012), i.e., the apparent temperature-dependence of mobilization of bioavailable P from soils.**

Referee comment: 8) Page 11, line 23: Perhaps mention threshold elemental ratios here, as well as the work that has focused on them in bacteria (Sinsabaugh, Chrzanowski, etc).

**Authors' comment: We have accepted the suggestions from Referee one (see Referee comment 1, 13, 15) and removed the discussion on threshold element ratios and inferences on C limitation. As the section in question has been deleted, we did not include these references.**

Referee technical corrections: 9 – 35)

**Authors' comment: We thank the Referee for technical corrections. All have been addressed in the new manuscript version, apart from the technical corrections in sentences that have been removed from the manuscript.**

36) Fig. 1: Why are the data points from the different treatments not differentiated here?

**Authors' comment: The majority of the data points overlap, differentiating these points would make the figure more complex and not necessarily more informative.**

**References:**
**Jansson, M., Berggren, M., Laudon, H., and Jonsson, A.: Bioavailable phosphorus in humic headwater streams in boreal Sweden, Limnology and Oceanography, 57, 1161-1170, 2012.**
**Stepanauskas, R., Leonardson, L., and Tranvik, L. J.: Bioavailability of wetland-derived DON to freshwater and marine bacterioplankton, Limnology and Oceanography, 44, 1477-1485, 1999.**
**Tranvik, L. J., and Hofle, M. G.: Bacterial growth in mixed cultures on dissolved organic carbon from humic and clear waters, Applied and Environmental Microbiology, 53, 482-488, 1987.**

---

## Author Response (AR1)

**New insights on resource stoichiometry: assessing availability of carbon, nitrogen and phosphorus to bacterioplankton**

Ana. R. A. Soares1, Ann-Kristin Bergström2, Ryan A. Sponseller2, Joanna M. Moberg1, Reiner Giesler4, Emma S. Kritzberg3, Mats Jansson2, Martin Berggren1

[revised manuscript text omitted]
. NH4NO3 was added to concentrations of 105, 133, 205, 305, 405  $\mu$ g N L-1, and Na2HPO4 was added to concentrations of 15.5, 18.8, 20.5, 30.5, 40.5  $\mu$ g P L-1 (see supplementary material Table 1). Standard curves for the rivers were based on the same approach method but bacterial responses to each concentration were recorded one time.

Integrated (cumulative) amounts of leucine incorporated by bacteria during lake or river bioassays over seven days were converted to concentrations of bioavailable element based on the slopes of the standard growth curves of either rivers or lakes, which describe how much leucine was incorporated per unit of bioavailable limiting element. For this conversion, the amount of incorporated leucine (given in nmol of leucine L-1 per-for seven days) during each bioassay was divided by the slope of the standard growth curve (nmol of leucine L-1 per mg of bioavailable nutrient L-1 for seven days). The resulting quotient represents the total amount of bioavailable nutrient taken up by bacterioplankton (mg L-1 for seven days; see supplementary material Table 3).).

**2.3 Leucine incorporation**

10

Measurements of protein synthesis were done using the method described by Smith and Azam (1992) and modified by Karlsson et al. (2002). Accordingly, 3H-leucine was added to sample water in Eppendorf tubes (specific activity varied between 60.5-115.8 Ci mmol-1, Perkin Elmer) to a final concentration of 30-100 nmol L-1. Additions of 3H-leucine were dependent on bacterial activity tests performed prior to the experiments where different concentrations of 3H-leucine identified the isotope saturation levels. Triplicate measurements were taken after 24 h, 48 h (we obtained six replicates at this time point), 72 h, 96 h and 168 h. Leucine incorporation into protein was determined by incubation for 1 h in the dark at 20 °C and incubations were terminated with trichloroacetic acid (TCA) additions of 5 % (w/v). A bacterial pellet was formed by centrifugation for 10 min at 14 000 rpm. The bacterial pellet was rinsed with 5 % TCA. After addition of 1.2 mL of scintillation cocktail (PerkinElmer) radioactivity was measured on a Wallac WinSpectral 1414 Scintillation counter (PerkinElmer). Incorporation of 3H-leucine was calculated using an intracellular dilution factor of 2 (Smith and Azam, 1992). Leucine incorporation measurements were integrated for the six time points and summed into a single value that represented the total amount of leucine incorporated for the seven\_day period. Lastly, at time point 96 h, an extra vial was collected and used as a blank, pre-treated with TCA 5\_% (w/v), followed by addition of leucine at a final concentration of 30 nmol L-1.

**2.4 Validation**

30

We validated the bacterial responses (leucine uptake) response to added amounts of BDOC, BDN and BDP (i.e., the slope of the standard cuves) by measuring relating the measured leucine uptake per unit ambient alternative estimates of bioavailable resources measured obtained with alternative independent methods. An alternative estimate of BDOC was obtained from measuring bacterial respiration (BR) during a lability incubation, which has been often applied in previous

aquatic research studies (del Giorgio and Cole, 1998; Jansson et al., 2000). The BR was determined by assessing decreases in dissolved oxygen concentrations from in bioassays water samples from lakes (n=13) and rivers (n=8). Sample water was prepared in parallel with, and in the same way as, the C bioassays described above. Volumes of 0.5 L were added to glass incubation bottles (in duplicate) which had sensors spots affixed to the inside surface. Oxygen concentrations were measured in the dark every 5 min for up to seven days with a FIBOX 3 (PreSens) that took optical readings from the outside of bioassay bottles. Estimates of BR were calculated from the averaged consumption of dissolved oxygen from the duplicate bottles by assuming a respiratory quotient of 1, which is a conservative value for unproductive lakes (Berggren et al., 2012). Bioavailable N was assessed using an alternative method described by Stepanauskas et al. (2000) by counting the cells produced in growth bioassays with N-starved bacteria. For this test, two aliquots of 30 mL were used for bioassays and one of them was amended with N-NH4NO3 to a final concentration of 0.405 mg N L-1. Both incubations were performed at 20 °C degrees in the dark. Bacterial biomass was determined at the start of the incubation (t=0) and after three days (t=3) after when the bacterial growth had peaked (Fig. 1). Bacterial samples were fixed with 3 % (v/v) glutaraldehyde and kept at 5 °C until analysis. Analyses of bacterial cells were conducted on a flow cytometer (FACScan, Becton Dickinson) on samples stained with SYTO 13 and run with addition of beads as internal standard according to del Giorgio et al. (1996), using CellQuest Pro software. Bacterial cells were distinguished based on green fluorescence intensity and side scatter signals. Total bacterial abundance was calculated as the sum of the populations that were distinguished in the cytograms. The N content per bacterial cell was determined by dividing the amount of N added to the amended aliquot by the difference in bacterial abundance between the N-amended and the unamended aliquot. To obtain BDN; the calculated average N content per cell was multiplied by the number of bacterial cells that were produced in the bioassay without addition. A more accurate method was used to To validate our We validated our estimates of leucine incorporation per unit bioavailable P by comparing it with the corresponding ratio in a completely independent boreal data set bioavailability using a more accurate approach that directly measures P accumulation in bacterial cells (Jansson et al. 2012). This independent data come from a freshwater study with near-identical bioassay conditions as in our P bioassays, with the major difference being that Jansson et al. (2012) used larger incubation volumes (> 700 mL) than we did when incubating in 1.5 mL Eppendorf tubes. Moreover, bioavailable P in the validation data was not assessed from bacterial growth data, but instead measured as P accumulation in bacterial cells harvested on filters. Such an approach This is possible for P7 since because standard TP instrumentation methods allows to measure changes in P concentration with provide high analytical precision at the microgram level (molybdenum blue method) and that can thus resolve small changes in P concentration. To do this Thus, www extracted the raw data from Jansson et al. (2012), where both cumulative leucine incorporation and bioavailability bioavailable P was were assessed by an alternative approach from quantified during incubation of water from two northern Swedish streams sampled on six dates from late April to late October 2010. , and in addition cumulative leucine incorporation during the bioassays was measured through the method described in this study. Hence, Here, This alternative approach was used to determine in this study concentrations of bioavailable P was determined 
[revised manuscript text omitted]
 are not not intended to representative of the long-term bioavailable pool in natural waters, but instead should be rather interpreted as bioavailable macro-elementnutrient estimates bioavailability across these very different sites did not seem to impact on the results determined under our specific laboratory conditions. The general pattern that we found across all sites was a relatively low bioavailability of C relative to that of N and P. This may suggest that C is more important as limiting factor for bacterial metabolism than previously thought. However, while Qour results can not be directly inform on the maximal pools of bioavailable macronutrients that can be readily consumed, the true exploitation of these resources in nature is transferred to natural systems dependent ason other (extrinsic) factors such as micro-element limitation, element co-limitation, and grazing pressures may also influence potential element bioavailability. Thus, based on our result alone it is not possible to determine whether or not the in situ bacterial metabolism was limited by a specific macronutrient, although it appears more likely that C would be limiting than N or P.

**4.4 Measuring bioavailability of C, N and P with leucine incorporation**

10

15

20

25

30

The linear relationships obtained from standard growth curves relating leucine incorporation to bioavailable resource concentrations showed that incorporation over a 7-day period was significantly and positively related to the amount of resource added. The fact that these relationships were not statistically different among the lakes suggests that leucine incorporation was driven by the added resources rather than other factors that could have affected the experiment. For example, variations in lake pH could have impacted the amount of resources taken up in the bioassays (del Giorgio and Davis 2003; Li et al., 2012). Because different methods were used, wDue to the lack of replication of We didcould not check

[revised manuscript text omitted]

|           |              | (143)              | ab                 | (64)               | (60)                     | (39)               | (22)                      | (21)                  |
| Lillsjöli | den          | 471                | 552 ab  | 334 cde | 176 ac        | 205 d   | 215                       | 176 be     |
| a         |              | (435)              | (338)              | (49)               | (36)                     | (7)                | (17)                      | (36)                  |
| Struptjä  | irnen        | 361 a   |                    | 432 b   | 692 acd       | 337 e   | 178 c          | 107 bde    |
| I a       |              | (327)              | 40 ob              | (93)               | (85)                     | (27)               | (21)                      | (6)                   |
| Stortjär  | nen          | 319 a   | 428 b   | 283°               | 301 d         | 213 e   | 104 abcdf      | 406 ef     |
|           |              | (210)              | (228)              | (49)               | (35)                     | (15)               | (8)                       | (130)                 |
|           |              |                    | į                  | b) BDN, μg I       | $NL^{-1}$                |                    |                           |                       |
| Övre      | Björntjärnen | 209 ab  |                    | 74°                | 61 a          | 84 d    | 73 e           | 23 bcde    |
|           |              | (13)               | 1.0                | (13)               | (6)                      | (14)               | (5)                       | (1)                   |
| Lillsjöli | den          | 287 abc | $232^{\text{def}}$ | 111 gh  | $33^{adg}$               | 64 be   | 89 a           | 51 cfh     |
|           |              | (10)               | (24)               | (7)                | (10)                     | (5)                | (6)                       | (3)                   |
| Struptjä  | irnen        | 259 abc |                    | 107 ad  | 220 e         | 273 dfg | 60 bf          | 37 ceg     |
| a         |              | (6)                | 20 ccef            | (28)               | (14)
67 be | 28                 | (6)
28 acfg | (2)                   |
| Stortjär  | nen          | 188 ab  | 206 cef | 82 ac   |                          | 84 f    | -                         | 119 g      |
|           |              | (15)               | (18)               | (6)                | (5)                      | (5)                | (3)                       | (38)                  |
|           |              |                    | •                  | c) BDP, μg I       | $PL^{-1}$                |                    |                           |                       |
| Övre      | Björntjärnen | 9 abc   |                    | 5 ad    | 5 be          | $9^{\mathrm{def}}$ | $7^{\mathrm{g}}$          | $3^{\rm cfg}$         |
|           |              | (1)                |                    | (0)                | (0)                      | (1)                | (0)                       | (0)                   |
| Lillsjöli | den          | $3^{ab}$           | 3°                 | $2^{def}$          | $2^{\text{cgh}}$         | 10 acd  | 7 beg          | $\hat{6}^{fh}$        |
|           |              | (0)                | (0)                | (0)                | (0)                      | (2)                | (1)                       | (0)                   |
| Struptjä  | irnen        | 6 ab    |                    | 6°                 | 9 ad          | 16 bce  | 7 e            | 4. de      |
| a         |              | (1)                | ocdef              | (0)                | (1)                      | (2)                | (1)                       | (0)
5 f |
| Stortjär  | nen          | O ab    | O cdef  | 1 gh               | 10 acg        | 12 bdh  | 6 e            |                       |
|           |              | (0)                | (0)                | (0)                | (2)                      | (2)                | (0)                       | (2)                   |

Table 3. Resource bioavailability in relation to the total resource pool, shown as percent bioavailable dissolved organic carbon (BDOC), bioavailable dissolved nitrogen (BDN) and bioavailable dissolved phosphorus (BDP). The data is divided into two groups which show average results for rivers with more than 10 mg C L-1 (rivers>10 mg C L-1; n = 3) and rivers with less than 10 mg C L-1 (rivers<10 mg C L-1; n = 4). Average element ratios of carbon to nitrogen (C:N), carbon to phosphorus (C:P), nitrogen to phosphorus (N:P) are calculated in molar for total (tot) and bioavailable resource fractions (bio). Ratios of dissolved inorganic nitrogen to phosphate (DIN:PO4-P) are also provided. Standard deviations are given within parentheses.

| Variable                | rivers >10 mg C L -1 | rivers <10 mg C L -1 |
|-------------------------|------------------------------------|------------------------------------|
| BDOC (%)                | 2 (1)                              | 3 (2)                              |
| BDN (%)                 | 48 (16)                            | 36 (20)                            |
| BDP (%)                 | 20 (12)                            | 31 (45)                            |
| C:N (bio)               | 1 (1)                              | 2(1)                               |
| C:N (total)             | 26 (5)                             | 24 (13)                            |
| C:P (bio)               | 319 (287)                          | 523 (795)                          |
| C:P (total)             | 1722 (378)                         | 920 (93)                           |
| N:P (bio)               | 294 (353)                          | 240 (251)                          |
| N:P (tot)               | 70 (27)                            | 46 (21)                            |
| DIN: PO 4 -P | 88 (68)                            | 2 (2)                              |

**Figure caption**

Figure 1: Leucine incorporation rates over the incubation time for a blank incubation and five spikes of C (spike 1=330, spike 2=660, spike 3=1000, spike 4=1330 and spike 5=1500  $\mu$ g C L-1), N (spike 1=105, spike 2=133, spike 3=205, spike 4=305 and spike 5=405  $\mu$ g N L-1) and P (blank, spike 1=15.5, spike 2=18.8, spike 3=20.5, spike 4=30.5 and spike 5=40.5  $\mu$ g P L-1).

5

Figure 2: Measurements of leucine incorporation in relation to additions of bioavailable C (as  $C_6H_{12}O_6$ ), N (NH4NO3) and P (Na2HPO4). Regression equations for all points pooled together: bioavailable C= 784x + 384 ( $R^2$  = 0.74, p

Figure 1.

Figure 2.

Figure 3.

Figure 4.

Figure 5.

We thank Referee 1 for constructive and relevant comments to the manuscript and for helping us to improve it. We addressed all comments below.

General comment Referee 1:" The manuscript by Soares and others is a novel and important contribution to this topic. In particular, their innovative experimental approach offers an answer to the question: what resource stoichiometry to bacteria actually experience in situ, given that not all measurable forms are bio-available? The work was thoughtfully designed and executed and will be of interest to the readership of Biogeosciences.

Two areas require attention from the authors. First, the conclusion that C is limiting is not adequately supported by the manuscript in its present form (see below). Second, the uncertainties in bioavailable concentrations must be made more clear. Aside from

20 these two areas, the paper is strong and the other comments are minor/clarification."

Referee comment 1:" Page 1 Line 24. What is the evidence for this in the present study? Although the resource stoichiometry derived from their results suggests that C will likely be limiting before N or P, this does not automatically mean that C is limiting. That extension of

resource stoichiometry is applicable only if 1) the bacteria are resource-limited and not under top-down control; 2) the only potentially limiting resources are C, N, or P; and 3) the system is presumed to be at steady state resembling a chemostat."

Authors' comment: We agree with the Referee and acknowledge that we do not present direct evidence showing C limitation. We have therefore reformulated all sentences in this regard, clarifying that C was the least bioavailable element out of the three key macronutrients that we work with. We also have now made clear that our bioavailability estimates are informative of maximum potential bioavailability under specific conditions, i.e. when all other macroand micronutrients of relevance are in excess. Thus, while we can state that access to bioavailable C in our samples tended to be in scarcity relative to the microbial need and access to N and P, the apparent C limitation is not directly transferrable to natural systems, especially not when considering the dynamic nature of natural ecosystems and the potential presence of top-down controls and/or micronutrient limitation.

40 Referee comment 2: Page 3 Line 8. While the long incubations have their shortcomings, it is overstated and confusing to say that these are not 'ecologically relevant timescales'. Certainly the majority of the consumption and respiration in fresh DOM happen in a matter of hours to days. However, longer-term degradation rates of more recalcitrant forms are of key importance. Specific to this study, the rapid rates of consumption observed are due to

the high concentrations of CNP added and thus, the timescale of the experiment is not ecologically relevant. I suggest that the authors focus this section and justification on the multi-element aspect of their design, which is the important and novel part.

25

30

35

Authors' comment: We agree with the Referee on the ecological importance of long-term degradation of more recalcitrant DOM, particularly in systems with long water residence times. However, resource bioavailability measured with long-term incubations does not reflect readily bioavailable pool sizes that control bacterial metabolism at a given moment. Moreover, during long incubation periods various factors can interfere with the uptake of bioavailable resources. For example, the dynamics of viruses and the development of toxic conditions that can appear from repeated bacterial regeneration of resources can interfere in long-term measurements (Cho et al., 1996). By using our seven-day approach and by maximizing bacterial metabolism, we reduce the incubation length to a minimum and sufficient time period during which bacteria take up most of the readily bioavailable pool (Fig. 1). Our estimates can be used to understand the potential C, N and P bioavailability, as they are performed during "ecologically relevant timescales". In our revised manuscript we will clarify that the relevance refers to how meaningful the measurements are for understanding the direct controls of bioavailable nutrient pools on the metabolism – not the controls the nutrient pools may have months ahead in time.

Referee comment 3: Page 3 Line 30. The third question seems certain to be true, and thus not informative as a question or hypothesis. Yet, quantifying this mismatch is important, so I suggest that the authors rewrite these questions.

Authors' comment: The third question was changed to "By how much do total C:N, C:P and N:P ratios exceed bioavailable C:N, C:P and N:P ratios".

Referee comment 4: Page 4 Line 10. By sampling the rivers at their outlet, much of the bioavailable forms have presumably been consumed in transit. What is the rationale for sampling far downstream from the sources of DOM?

Authors' comment: Our main goal was to capture bioavailability patterns across a landscape gradient with different boreal freshwater properties (please see revised manuscript version page 4 lines 7-9) and not to determine the amount of bioavailable element coming from terrestrial soils.

Referee comment 5: Page 5 Line 2. This standardized inoculum has important implications for interpreting the results. Elaborate on why this single community was used as opposed to the communities present in the source water.

Authors' comment: We wanted to ensure that differences in bacterial community composition did not influence our estimates of resource bioavailability (Martinez et al., 1996). This was achieved by using a standard bacterial community in all our assays. We have now explicitly motivated the use of a singular bacterial inoculum in the manuscript. By using a pooled inoculum representing both headwater inlet and lake water from four different lakes with different properties, we ensured a high diversity of the microbial assemblage that was used to inoculate.

Referee comment 6: "Page 5 Line 15-30. This experimental approach is rather involved. If space allows, the authors should include a schematic diagram that shows how they forced limitation by CNP and measured the response to addition of the limiting resource. Presumably this method is based on the Wright-Hobbie technique and thus it is important to show how the estimates of ambient concentrations were derived."

Authors' comment: We agree that it is important to include a schematic diagram to help to better visualize our approach. We will add a schematic diagram of the method to the supplementary material in our next manuscript version.

Referee comment 7: "Page 5 Line 30. "The total amount of bioavailable nutrient taken up" is not precise. Especially for C, the nutrient need not be assimilated in order for the bacteria to exhibit a growth response."

Authors' comment: We think that our sentence is well formulated. We used leucine incorporation as an experimental response variable of all bioavailable element uptake, which in the case of C can be used either for growth or respiration.

Referee comment 8: "Page 6 Line 15. The use of complementary validation methods is an important strength of this paper. Well done."

Author's comment: Thank you for pointing this out.

25

30

35

Referee comment 9: "Page 6 Line 32. This method of calculating cellular N content is strange. What are the assumptions of this method? At the least it assumes that all of the added N is assimilated and that no other N is used."

Authors' comment: This method encompasses several assumptions: 1) bacterial growth in the bioassays was effectively limited by N, 2) different N compounds yield similar bacterial biomass increases, 3) all bioavailable N was assimilated when bacterial growth ceased and 4) N bioavailability was independent from the bacterial inocula. The paper from Stepanauskas et al. (1999) describes in detail the experimental setup and the method's assumptions.

5

10

15

Referee comment 10: "Page 7 Line 5. The validation method used for P availability is more straightforward than for N. Why not use this method for N also? Additionally, were these fitler-P measurements corrected/checked for phosphate binding to the filter?"

Authors' comment: We lacked the equipment necessary to measure bioavailable N (and C) with the same method as the one used to determine P bioavailability.

Estimates of P bioavailability were corrected for potential P filter content, binding of dissolved P species and abiotic formation of particles (Jansson et al., 2012).

20

Referee comment 11: "Page 7 Line 30. Needs clarification. No difference between slopes for C, N, and P or among lakes? Also, it is unclear why the regressions were performed individually for each analytical replicate instead of using all of the analytical replicates for a given site/date. From what I can tell, the standard curves were computed individually for each of five analytical replicates and then the standard deviation of their estimates is presented in table 2?"

Authors' comment: We have changed lines 24-25 on page 8 of the revised manuscript.

25

We first performed the regressions individually (Figure 2), precisely because we wanted to test whether there were differences on the bacterial response to nutrient additions between the different lakes. Since we found no statistically significant differences between lake slopes (this is mentioned on page 8 line 24 and page 13 lines 30-31 of the revised manuscript), we combined all datapoints and performed a new regression for each element based the entire dataset. This rendered the "mean slope" given on Figure 2 (C slope=784 nmol  $L^{-1}$  per  $\mu$ g C  $L^{-1}$ , N=slope 2667  $\mu$ g N  $L^{-1}$ , P slope=67575  $\mu$ g P  $L^{-1}$ ).

No, in table 2, the mean slope of the standard curves was used to translate amounts five replicate measurements of leucine uptake. The standard deviation of the estimates is given within brackets.

Referee comment 12: "Page 9 Line 20. Were the total and bioavailable concentrations (or elemental ratios) positively correlated?"

Authors' comment: No, there was no correlation between the total and bioavailable concentrations.

5

30

Referee comment 13: "Page 9 Line 23. Again, what is the evidence that C was most limiting, or even limiting at all? The traditional lines of evidence for this (single nutrient bioassays) are not presented, so this is either inferred from the stoichiometry estimated for resources or

from the low proportional bioavailability of C compared to N and P. Neither of these shows that C was the strongest limiting factor. Pease elaborate on this and explain 1) the assumptions used for this claim and 2) the specific evidence from this study"

Authors' comment: We agree with the Referee that we do not have the evidence needed to claim that C is limiting in boreal waters (see answer to Referee comment 1). We have changed the sentence in question (see revised manuscript page 10, lines 17-18).

"Surprisingly, in these systems where absolute surface water DOC concentrations are large, C bioavailability was low and was the strongest limiting factor for heterotrophic aquatic production." to "In these systems where absolute surface water DOC concentrations are large, C bioavailability was lowest, relative to N and P."

Referee comment 14: Page 10 Line 33. There are many other factors related to seasonality that could explain this (light, plant production, hydrology, etc), so how can you conclude that soil microbial activity is the predominant driver? Overall, I found this discussion of seasonality too speculative

Authors' comment: We acknowledge the important role of other seasonal factors for the amount of bioavailable dissolved organic carbon measured in our study. We have now removed the sentences from the revised manuscript page 11 lines 26-31.

Referee comment 15: Page 11 Line 27. These calculations seem to be the core of the argument that C is limiting and thus require elaboration. Even then, this only shows that C is more likely to be limiting than N or P, but does not show that C was in fact limiting at ambient concentrations.

Moreover, the ranges here are so large that they are not really meaningful. Why not use the ratio of slopes presented in figure 2 to estimate the relative consumption rates

of CNP? In your calculations, you already assume that the ratio of leucine:cell is invariant, so the ratio of 1/C-slope to 1/P-slope (=86) is the ratio of C consumption to P consumption when those elements are limiting. No?

In both the lakes and the rivers, the DOM pools have already undergone much degradation by bacteria, light, and reactive oxygen. This needs to be acknowledged, or better yet, discussed in some detail.

Authors' comment: We agree with the Referee. We have thus, reformulated our conclusion and all statements related to the topic (please see also answers to Referee comment 1 and 13).

We decided to exclude the calculations from the manuscript, as we acknowledge that the use of natural ranges of BGE may not be truly representative of BGE values in our bioassays in which element limitation was strongly induced. We appreciate the reviewer's suggestion of assuming a ratio for the C consumption in relation to P but we now think it is better to remove the discussion of C limitation of BP.

We agree with the Referee regarding the loss of most of the riverine bioavailable pool. We added a discussion paragraph on the subject to the revised manuscript version (page 13 lines 9-19). This however does not apply to the lakes DOM as we targeted the short-term bioavailable resource pool (see revised manuscript version page 13 lines 9-19).

20

15

Referee comment 16: "Page 13, line 1. Avoiding these uncertainties is important, but those are typically on the order of a few percent and can be constrained by experimental validation. Without a robust analysis of the resulting uncertainties from the present approach, it is not

25 possible to discern which method is advantageous. Form Table 2 and Figure 1/2, it appears that the uncertainty in concentration estimated for a single date/site is large. Without such an analysis of the uncertainty in the final estimates, I suggest that the authors focus on the multi-element aspects of their study"

- Authors' comment: As suggested we will focus our discussion on the multi-element aspect of our study. Thus, we have removed from the revised manuscript lines 21 to 25 on page 14.
- 35 Referee comment 17: "Figure 4. What do the diamonds represent in this figure?"

Authors' comment: The diamonds represent average resource ratio values for the lakes for all dates (n=26). We have added to Figure's 4 caption the following sentence: "Data shown as boxplots and includes mean as diamonds.

Referee comment 18: "Figure 5. The vertical axis scale should be fitted to the range of data presented."

**5 Authors' comment: Vertical axis scale has been changed from 1 to 100000 to 100 to 100000.**

We thank referee 2 for constructive and relevant comments and suggestions of technical corrections, which helped us to improve the manuscript. Please find our response below.

10

**General Comments**

This manuscript presents the results from a test of a new method of determining the relative bioavailability of carbon, nitrogen and phosphorus for lake and riverine bacterioplankton. The technique, which combines radiolabeled leucine incubations with reciprocal nutrient amendments, is a novel approach to backing out the proportion of total dissolved C, N and P that bacteria can rapidly take up if other factors are not limiting. The authors test the approach with seasonal samples from four Swedish lakes and single-date samples from seven rivers. Overall, the authors provide a very interesting analysis and the paper is in good shape. Please see below for my specific and technical comments. The only general comment that I would make is that the approach explicitly considers bioavailability in the absence of any co-limitation. In other words, the method cannot incorporate any interactions between limiting factors. While this may be a necessary shortfall of the approach, its significance perhaps deserves some thought and maybe some treatment in the discussion.

Authors' comment to the general comment: The Referee is correct. Our method determines the maximum pool sizes of readily bioavailable macronutrient fractions that can be used given that all other nutrients are provided in access. In the revised manuscript version we clarify that our bioavailability estimates are defined under these specific operational conditions. We also explain that, in order to translate the implications of the results to natural systems, factors like nutrient co-limitation and potential limitation by micronutrients or even top-down controls (e.g., grazing as pointed out by Reviewer 1) need to be taken into account (see revised manuscript version page 13 lines 21-26).

30

45

**Specific Comments**

Referee comment: 1) "Page 1, line 17-18: Make sure the readers know that these percentages are based on the initial concentrations. I know that this might sound obvious, but I was initially confused about whether these were percentages of final (post-incubation) or initial (pre-incubation) amounts."

Authors' comment: We thank the Reviewer for pointing this out. This has been clarified in the new manuscript version (page 1, lines 16-18).

Referee comment: 2) "Page 5, line 2: Where exactly was the inoculum sampled? And how could it have been sampled only once, given that the lake and river samples were collected over a lengthy period and the incubations run soon after each sample collection? Was it maintained in the laboratory?"

Authors' comment: The inoculum consisted of a mixture of water from both the epilimnion and inlet of the lakes. By including communities from several different sampling sites, we ensured a large microbial diversity on the inoculum. The inoculum was maintained in the fridge at approximately 4 °C. Because our experiment strongly maximized bacterial metabolism (selecting for fast-growing opportunistic bacteria), we do not think that the inoculum played a large role on the outcome of our experiment. Previous studies have further demonstrated bacterial growth to be independent of bacteria inocula (Tranvik and Hofle, 1987).

Referee comment: 3) Page 5, line 8: Could there be an effect of incubating bacterioplankton in such a small volume of water? Could biofilms on the walls of these small vials start to have a disproportionate impact on the results?

Authors' comment: We did not test or control for the potential development of biofilms in the tubes walls. However, based on the results for phosphorus presented in Figure 5, we can compare our measurement of the amount of leucine incorporation (normalized per unit of bioavailable P; filled square) with corresponding data extracted from Jansson et al. (2012; the box plot). In the latter case, Jansson et al. did not involve incubations in Eppendorf tubes but in much larger (700+ ml) volumes. There was an overlap in magnitude of leucine incorporation when comparing these two data sources, but it can be noted that our measurements are in the upper range compared to those from Jansson et al. (2012). Biofilm accumulation could have potentially contributed to this difference in our incubation tubes. However, when looking at the time series of our incubations (Figure 1), it is clear that most of the leucine incorporation in our case happened already within 3 days, which should be a time-frame too short for substantial biofilm formation. Thus, we do not consider that biofilms strongly influenced our results.

Referee comment: 4) Page 5, line 16: Maybe I'm missing something, but why didn't the controls consist of lake water without any added C, N or P?

25

50

Authors' comment: Since our design is based on the idea of inducing strong limitation of the nutrient to be evaluated for maximum potential bioavailability, we did not consider relevant to incubate lake water without any nutrient additions. On the lines that the Reviewer refers to, we tested whether the inoculum or L16 added any bioavailable C, N and P to our assays. By using Mili-Q water instead of lake water, we made sure the inoculum and L16 were the only possible sources of limiting resource in our bioassays. At the same time, this also tested that leucine incorporation (or bacterial growth) was in fact controlled by the induced limiting resource and that no bacterial growth occurred in the absence of the bioavailable limiting resource (see revised manuscript version page 14 lines 3-5).

40 Referee comment: 5) Page 5, line 24: Presumably these standard curves would be system-specific? Or at least limited to similar environments within a region? Some discussion of should perhaps be added to the discussion.

Authors' comment: We did not find significant differences among standard curves for the different lakes (page 13, line 6 revised manuscript version), which is interesting since the lakes represent gradients in DOC and catchment features representative to a range of boreal conditions. Possibly, corresponding standard curves from the rivers could have been different from those in the lakes, but it would have been a major time-consuming effort to determine those curves for all of the rivers. Therefore, results from the rivers should be interpreted with caution, even if the rivers do not represent fundamentally different chemical conditions. We added a short section on the subject to the discussion part in the new manuscript (pages 13-14 lines 33-3).

Referee comment: 6) Page 6, line 24 to page 7, line 8: It sounds like these methods assume no changes in cellular stoichiometry with nutrient availability (i.e. elemental homeostasis).

Authors' comment: Yes the Reviewer is correct; an invariant cellular stoichiometry was assumed in the validation method used to calculate N bioavailability. This, as well as other method assumptions, has been scrutinized in Stepanauskas et al. (1999).

10 However, there were no assumptions regarding cellular stoichiometry for the method used to calculate P bioavailability since this was based on direct measurements of the content of P in bacterial growth cultures harvested from filters.

Referee comment: "7) Page 11, line 5: Is this consistent with turnover rates of these elements in these ecosystems?"

Authors' comment: We do not know of studies looking at the turnover rates of these elements in the soils in the study area. We have deleted the sentence in question and replaced it with a clearer sentence that brings into attention the main mechanism suggested by Jansson et al. (2012), i.e., the apparent temperature-dependence of mobilization of bioavailable P from soils (see revised manuscript page 11-12, lines 33-3).

Referee comment: 8) Page 11, line 23: Perhaps mention threshold elemental ratios here, as well as the work that has focused on them in bacteria (Sinsabaugh, Chrzanowski, etc).

Authors' comment: We have accepted the suggestions from Referee one (see Referee comment 1, 13, 15) and removed the discussion on threshold element ratios and inferences on C limitation. As the section in question has been deleted, we did not include these references.

Referee technical corrections: 9 - 35)

Authors' comment: We thank the Referee for technical corrections. All have been addressed in the new manuscript version, apart from the technical corrections in sentences that have been removed from the manuscript.

36) Fig. 1: Why are the data points from the different treatments not differentiated here?

Authors' comment: The majority of the data points overlap, differentiating these points would make the figure more complex and not necessarily more informative.